# SIMPLIFYING TRANSFORMER BLOCKS

**Bobby He & Thomas Hofmann**[*]
Department of Computer Science, ETH Zurich

## ABSTRACT

A simple design recipe for deep Transformers is to compose identical building blocks. But standard transformer blocks are far from simple, interweaving attention and MLP sub-blocks with skip connections & normalisation layers in precise arrangements. This complexity leads to brittle architectures, where seemingly minor changes can significantly reduce training speed, or render models untrainable.

In this work, we ask if the standard transformer block can be simplified? Combining signal propagation theory and empirical observations, we motivate modifications that allow many block components to be removed with no loss of training speed, including skip connections, projection or value parameters, sequential sub-blocks and normalisation layers. In experiments on both autoregressive decoder-only and BERT encoder-only models, our simplified transformers emulate the per-update convergence speed and performance of standard transformers, while enjoying 16% faster training throughput, & using 15% fewer parameters.

## 1 INTRODUCTION

The transformer architecture (Vaswani et al., 2017) is arguably the workhorse behind many recent successes in deep learning. A simple way to construct a deep transformer architecture is by stacking multiple identical transformer "blocks" one after another in sequence. Each block, however, is more complicated and consists of many different components, which need to be combined in specific arrangements in order to achieve good performance. Surprisingly, the base transformer block has changed very little since its inception, despite attracting the interest of many researchers.

In this work, we study whether the standard transformer block can be simplified. More specifically, we probe the necessity of several block components, including skip connections, projection/value matrices, sequential sub-blocks and normalisation layers. For each considered component, we ask if it can be removed without loss of training speed (both in terms of per-update step & runtime), and what architectural modifications need to be made to the transformer block in order to do so.

We believe the problem of simplifying transformer blocks without compromising training speed is an interesting research question for several reasons. First, modern neural network (NN) architectures have complex designs with many components, and it is not clear the roles played by these different components in NN training dynamics, nor how they interact with each other. This is particularly pertinent given the existing gap between theory and practice in deep learning, where theorists working to understand the mechanisms of deep learning often only consider simplified architectures due to convenience, not necessarily reflective of modern architectures used in practice. Simplifying the NN architectures used in practice can help towards bridging this divide.

On a related theoretical note, our work highlights both strengths and current limitations of signal propagation: a theory that has proven influential due to its ability to motivate practical design choices in deep NN architectures. Signal propagation (Poole et al., 2016; Schoenholz et al., 2017; Hayou et al., 2019) studies the evolution of geometric information in an NN at initialisation, captured through inner products of layerwise representations across inputs, and has inspired many impressive results in training deep NNs (Xiao et al., 2018; Brock et al., 2021; Martens et al., 2021; Zaidi et al., 2023). However, the current theory only considers a model at initialisation, and often considers only the initial forward pass. As such, signal propagation at present is unable to shed light on many intricacies of deep NN training dynamics, for example the benefits of skip connections for training speed. Though signal propagation is crucial in motivating our modifications, we would not have arrived at our simplified transformer blocks from theory alone, and relied also on empirical insights.

---

[*]Correspondence to: `bobby.he@inf.ethz.ch`.

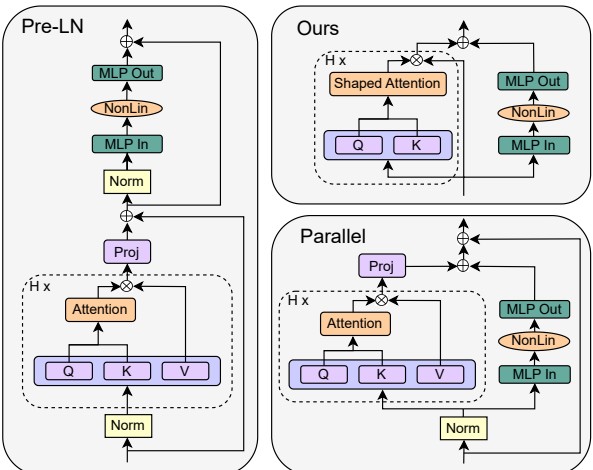

Figure 1: Comparison between different Transformer blocks. (Left) The standard Pre-LN block. (Top Right) Our most simplified block. (Bottom Right) The parallel block (Zhao et al., 2019; Wang & Komatsuzaki, 2021). Like the parallel block, our block eschews the need for sequential sub-blocks, but we additionally remove all skip connections and normalisation layers, as well as value and projection parameters. Here, $\otimes$ denotes a matrix multiplication, and $\oplus$ denotes a (potentially weighted) sum.

Finally, on the practical side, given the cost of training and deploying large transformer models nowadays, any efficiency gains in the training and inference pipelines for the transformer architecture represent significant potential savings. Simplifying the transformer block by removing non-essential components both reduces the parameter count and increases throughput in our models. In particular, we show that it is possible to remove skip connections, value parameters, projection parameters and sequential sub-blocks, all while matching the standard transformer in terms of training speed and downstream task performance. As a result, we reduce parameter count by up to 16% and observe throughput increases of 16% at both train and inference time.

Our starting point to simplify Transformer blocks is He et al. (2023), who show that respecting signal propagation principles allows one to train deep Transformers without skip connections or normalisation layers, but at significantly reduced convergence speeds per parameter update. We first show that regulating the updates to values and projection parameters (Sec. 4.1), or in fact removing them entirely (Sec. 4.2), improves the performance of skipless attention sub-blocks, and recovers the lost per-update training speed reported by He et al. (2023). This removes half of the parameters and matrix-multiplications in the attention sub-block. In Sec. 4.3, we show our simplifications combine profitably with parallel sub-blocks (Zhao et al., 2019; Wang & Komatsuzaki, 2021), allowing us to remove all remaining skip connections and sequential sub-blocks without compromising per-update training speed, whilst further boosting the throughput increase to be 16%, in our implementation. Finally, in Sec. 5, we show that our simplified blocks improve when scaled to larger depths, work well in both encoder-only and decoder-only architectures, and that our findings also hold when scaling training length. We conclude with a discussion of limitations and future work in Sec. 6.

## 2    RELATED WORK

Simplifying deep NNs by removing block components has received a lot of attention, both in transformers and other architectures. In these works, signal propagation theory often acts as inspiration.

For a pair of inputs $\boldsymbol{x}, \boldsymbol{x}'$, mapped to a pair of representation/activations vectors $\boldsymbol{x}_l, \boldsymbol{x}'_l \in \mathbb{R}^d$ at layer $l$, signal propagation theory studies the evolution of activation inner products $\frac{1}{d}\boldsymbol{x}_l^\top \boldsymbol{x}'_l, \frac{1}{d}\boldsymbol{x}_l^\top \boldsymbol{x}_l, \frac{1}{d}\boldsymbol{x}'_l{}^\top \boldsymbol{x}'_l$ at initialisation, which can be tracked with their large $d$ limits (Lee et al., 2018; Matthews et al., 2018; Yang, 2019). Several pathologies afflicting poorly designed deep NNs can be identified as a result (Schoenholz et al., 2017; Hayou et al., 2019; Yang et al., 2019; Dong et al., 2021; Martens et al., 2021). For example, the activation norms $\frac{1}{d}\boldsymbol{x}_l^\top \boldsymbol{x}_l$ may blow up or vanish, or the cross products $\frac{1}{d}\boldsymbol{x}_l^\top \boldsymbol{x}'_l$ may converge to a value independent of the inputs $\boldsymbol{x}, \boldsymbol{x}'$ at large $l$, in which case deeper layers of the model are unable to identify different inputs. Avoiding such degeneracies is important to allow for good training dynamics and generalisation in deep NNs (Balduzzi et al., 2017; Xiao et al., 2018; 2020; Hayou et al., 2021; Martens et al., 2021; Noci et al., 2022).

It has been shown that judicious use of weight initialisations and architectural tools, like skip connections and normalisation layers, can improve signal propagation degeneracies and the trainability of deep NNs. Such considerations have motivated principled modifications with simpler architectures. De & Smith (2020) show that an implicit mechanism of Pre-LN skip connections is to downweight the residual branch relative to the skip branch, leading to better signal propagation. They also show that explicitly downweighting the residual branch allows normalisation layers to be removed without affecting performance. The idea of downweighting residuals for improved signal propagation & trainability has been studied extensively in the literature (Zhang et al., 2018; Hanin & Rolnick, 2018; Tarnowski et al., 2019; Zhang et al., 2019; Arpit et al., 2019; Xu et al., 2020; Bachlechner et al., 2021; Touvron et al., 2021; Hayou et al., 2021; Hayou & Yang, 2023; Martens et al., 2021; Davis et al., 2021; Noci et al., 2022; Wang et al., 2022a; Huang et al., 2020; Wang et al., 2022b).

For skip connections (He et al., 2016), it has been shown that transforming non-linear activation functions in MLPs and CNNs to be more linear according to a given deep architecture can enable good signal propagation even without skip connections (Martens et al., 2021; Zhang et al., 2022; Li et al., 2022). He et al. (2023) apply similar considerations to the self-attention mechanism, where the key insight is that attention matrices need to be more identity-like in order to prevent signal degradation in skipless transformers. However, these works find that skipless architectures suffer from significant losses in training speed compared to their residual counterparts, when using standard optimisers like SGD or Adam. Such differences were not observed with stronger optimisers like K-FAC (Martens & Grosse, 2015) on CNNs, and this inability to explain training phenomena highlights a current limitation of signal propagation theory. Ding et al. (2021; 2023) design a CNN, RepVGG, that can be trained like a residual architecture for fast per-update convergence, but reparameterised to be skipless at test time for significantly higher inference throughput. This reparameterisation is related to our considerations of value and projection parameters in Sec. 4.

Many works have considered simplifications or improvements specific to the transformer. Most relevant to our work is the parallel block (Zhao et al., 2019; Wang & Komatsuzaki, 2021) (pictured Fig. 1, bottom right), which computes the MLP and attention sub-blocks in parallel for efficiency gains, with minimal performance loss. Trockman & Kolter (2023) observe that the product of value and projection parameters often has a large identity component in trained transformers, and design an initialisation mimicking this to improve performance in standard transformers on small datasets. We find these matrices can be fixed to the identity without loss of performance, which removes them from our simplified architecture. Other works have considered reducing the frequency of MLP sub-blocks (Sridhar et al., 2022; Pires et al., 2023) or efficient replacements to softmax attention (Katharopoulos et al., 2020; Schlag et al., 2021; Choromanski et al., 2021). Sukhbaatar et al. (2019) remove the MLP by integrating it into the attention sub-block, augmented with persistent memory.

## 3  PRELIMINARIES

A deep transformer architecture of depth $L$ is formed by sequentially stacking $L$ transformer blocks. The most common block is Pre-LN, depicted in Fig. 1 (left), which we treat as a baseline for comparing training speed, both in terms of per-update and runtime. It differs from the original Post-LN block only in the position of the normalisation layers relative to the skip connections, but is more popular as the Post-LN block suffers from poor training stability and signal propagation in deep layers (Xiong et al., 2020; Liu et al., 2020; Noci et al., 2022; He et al., 2023).

Transformer blocks take representations of sequences as inputs. For an input sequence representation $\mathbf{X}_{\text{in}} \in \mathbb{R}^{T \times d}$, with $T$ tokens and dimension $d$, the Pre-LN block outputs $\mathbf{X}_{\text{out}}$, where:

$$\mathbf{X}_{\text{out}} = \alpha_{\text{FF}}\, \hat{\mathbf{X}} + \beta_{\text{FF}}\, \text{MLP}(\text{Norm}(\hat{\mathbf{X}})), \quad \text{where } \hat{\mathbf{X}} = \alpha_{\text{SA}}\, \mathbf{X}_{\text{in}} + \beta_{\text{SA}}\, \text{MHA}(\text{Norm}(\mathbf{X}_{\text{in}})). \quad (1)$$

with scalar gain weights $\alpha_{\text{FF}}, \beta_{\text{FF}}, \alpha_{\text{SA}}, \beta_{\text{SA}}$ fixed to 1 by default. Here, "MHA" stands for Multi-Head Attention (detailed below), and "Norm" denotes a normalisation layer (Ba et al., 2016; Zhang & Sennrich, 2019). In words, we see that the Pre-LN transformer block consists of two sequential sub-blocks (one attention and one MLP), with normalisation layers and residual connections for both sub-blocks, and crucially the normalisation layers are placed within the residual branch. The MLP is usually single hidden-layer, with hidden dimension that is some multiple of $d$ (e.g. 4 (Vaswani et al., 2017) or 8/3 (Touvron et al., 2023)), and acts on each token in the sequence independently.

The MHA sub-block allows tokens to share information between one another using self-attention. For input sequence $\mathbf{X}$, the self-attention mechanism outputs:

$$\text{Attn}(\mathbf{X}) = \mathbf{A}(\mathbf{X})\mathbf{X}\mathbf{W}^V, \quad \text{where } \mathbf{A}(\mathbf{X}) = \text{Softmax}\left(\frac{1}{\sqrt{d_k}}\mathbf{X}\mathbf{W}^Q\mathbf{W}^{K\top}\mathbf{X}^\top + \mathbf{M}\right), \quad (2)$$

where $\mathbf{W}^Q, \mathbf{W}^K \in \mathbb{R}^{d \times d_k}$ and $\mathbf{W}^V \in \mathbb{R}^{d \times d_v}$ are trainable query, key and value parameters respectively. Here, the attention matrix $\mathbf{A}(\mathbf{X}) \in \mathbb{R}^{T \times T}$ can be thought of as allowing different tokens to "mix" with each other. $\mathbf{M} \in \mathbb{R}^{T \times T}$ is a mask taking values in $\{0, -\infty\}$ that depend on the modelling task. For causal auto-regressive transformers like GPT, $\mathbf{M}_{i,j} = 0$ iff $i \geq j$, which prevents a token from obtaining information from future tokens. In bidirectional models like BERT, masking is typically applied at the token level and not in the attention mechanism (i.e. $\mathbf{M}$ is the zero matrix).

The Multi-Head Attention name arises because it is typical in practice to apply self-attention on $H$ different "heads" (with independent parameters) with $d_v = d_k = \frac{d}{H}$, as follows:

$$\text{MHA}(\mathbf{X}) = \text{Concat}\big(\text{Attn}_1(\mathbf{X}), \ldots, \text{Attn}_H(\mathbf{X})\big)\mathbf{W}^P, \quad (3)$$

where $\mathbf{W}^P \in \mathbb{R}^{d \times d}$ denotes a trainable square projection matrix that combines different attention heads. If we let $\mathbf{W}_n^V$ denote the value parameters for head $n$, then the concatenated value weights $\mathbf{W}^V = \text{Concat}(\mathbf{W}_1^V, \ldots, \mathbf{W}_H^V) \in \mathbb{R}^{d \times d}$ can be viewed as a square matrix. One of our key findings, in Sec. 4.2, is to show that fixing the value and projection parameters, $\mathbf{W}^V$ and $\mathbf{W}^P$, to the identity matrix significantly improves per-update training speed in skipless transformer blocks (to speeds matching or even outperforming the standard Pre-LN block), whilst simultaneously significantly reducing the parameter count and matrix-multiplication FLOPs required, thus increasing throughput.

# 4  SIMPLIFYING TRANSFORMER BLOCKS

We now describe how we arrive at our simplest Transformer block, Fig. 1 (top right), starting from the Pre-LN block, using a combination of signal propagation theory and empirical observations. Each subsection here will remove one block component at a time without compromising training speed, and we aim to provide an intuitive account of our progress in simplifying the Pre-LN block.

All experiments in this section use an 18-block 768-width causal decoder-only GPT-like model on the CodeParrot dataset,[1] which is sufficiently large that we are in a single epoch regime with minimal generalisation gap (Fig. 2), allowing us to focus on training speed. We provide depth scaling, and non-causal encoder-only, experiments, in Sec. 5. We use a linear decay learning rate (LR) schedule[2] with AdamW (Loshchilov & Hutter, 2017), with linear warmup for the first 5% steps. The maximum LR is tuned on training loss, using a logarithmic grid. Additional experimental details are in App. D.

## 4.1  REMOVING THE ATTENTION SUB-BLOCK SKIP CONNECTION

We first consider a skipless attention sub-block, whose output has the simple interpretation of adding, to each token, other token representations according to the attention matrix. In the notation of Eq. (1) this corresponds to $\alpha_{\text{SA}} = 0$. Naively removing the attention skip leads to a signal degeneracy called rank collapse (Dong et al., 2021), which harms trainability (Noci et al., 2022).

**Setup**  He et al. (2023) outline modifications needed to the self-attention mechanism in order to correct these signal degeneracies at large depths, and train such deep skipless networks for the first time. One method they introduce, Value-SkipInit, modifies the self-attention matrix to compute:

$$\mathbf{A}(\mathbf{X}) \leftarrow (\alpha\mathbf{I}_T + \beta\mathbf{A}(\mathbf{X})) \quad (4)$$

with trainable scalars $\alpha, \beta$ initialised to 1 and 0 respectively, and $\mathbf{I}_T \in \mathbb{R}^{T \times T}$ is the identity matrix.

The key insight here is to initialise the self-attention matrix to have a dominant identity component that encourages a token to attend to itself more relative to other tokens, much in the same way that a Pre-LN skip upweights the skip branch relative to the residual branch for good signal propagation at large depths (De & Smith, 2020). We point out that these considerations only apply at initialisation.

Noci et al. (2023) propose an extension, Shaped Attention, also motivated by signal propagation:

$$\mathbf{A}(\mathbf{X}) \leftarrow (\alpha\mathbf{I}_T + \beta\mathbf{A}(\mathbf{X}) - \gamma C). \quad (5)$$

---

[1]Our setting is taken from `https://huggingface.co/learn/nlp-course/chapter7/6`.
[2]We found linear decay to slightly outperform cosine decay for both our models and baselines (c.f. Fig. 12).

Here, $\alpha, \beta, \gamma$ are trainable, and $C$ is a constant (not trained) centering matrix, set to be equal to the values of $\mathbf{A}$ when the query-key dot product $\frac{1}{\sqrt{d_k}}\mathbf{X}\mathbf{W}^Q\mathbf{W}^{K^\top}\mathbf{X}^\top$ is zero[3]. Like He et al. (2023), we initialise queries $\mathbf{W}^Q = 0$, which exactly zeros the query-key dot product at initialisation. Then, $\beta = \gamma$ means that $\beta\mathbf{A}(\mathbf{X}) - \gamma C = 0$ at initialisation, and $\alpha = 1$ ensures a dominant identity component, and good signal propagation. Ali et al. (2023) also centre attention and show it helps prevent oversmoothing in vision transformers and graph NNs.

We found Shaped Attention, Eq. (5), to slightly outperform Eq. (4) (c.f. Fig. 13), and use it in our experiments on skipless attention sub-blocks, with $\beta = \gamma = \alpha = 1$ at initialisation unless stated otherwise. We also use head-dependent scalars in Eq. (5), $\alpha_h, \beta_h$ and $\gamma_h$, which provided a small additional performance boost. One final important implementation detail is that for any skipless block we explicitly downweight the MLP branch by initialising trainable $\beta_{\text{FF}} = O(\frac{1}{\sqrt{L}}) < 1 = \alpha_{\text{FF}}$. This is motivated through signal propagation theory (c.f. Stable ResNet, Hayou et al. (2021)), and accounts for the fact that removing skip connections (in either MLP or

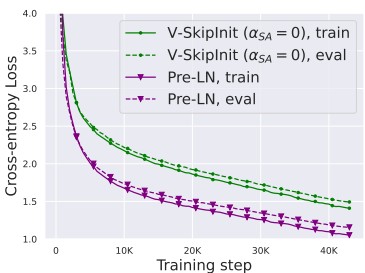

Figure 2: Loss of training speed in transformers without attention sub-block skip (He et al., 2023), even with Shaped Attention, Eq. (5), and MLP skips ($\alpha_{\text{FF}} = 1$).

MHA sub-block) reduces the implicit downweighting effect of Pre-LN blocks (De & Smith, 2020). For the depth $L = 18$ networks in this section, we initialise $\beta_{\text{FF}} = 0.1$.

**Recovering lost training speed** Despite allowing skipless transformers to train for the first time, He et al. (2023) reported a significant loss of training speed per step compared to the Pre-LN block. We verify this in Fig. 2.

To recover the lost training speed without attention skips, note that identity attention matrices make a deep transformer with no MLP sub-blocks act like a deep skipless linear NN at initialisation,[4] $f(\mathbf{X}) = \mathbf{X}\prod_{l=1}^{L}\left(\mathbf{W}_l^V\mathbf{W}_l^P\right)$, where $\mathbf{W}_l^V, \mathbf{W}_l^P$ are the value and projection weights in layer $l$. In He et al. (2023), they initialise $\mathbf{W}_l^V, \mathbf{W}_l^P$ to be independent random orthogonal matrices to avoid signal

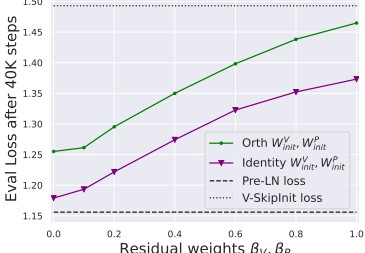

Figure 3: Restricting updates to $\mathbf{W}^V, \mathbf{W}^P$, through smaller $\beta_V, \beta_P$, recovers training speed in skipless transformers ($\alpha_{\text{SA}} = 0$).

degeneracies from Gaussian initialisations (Saxe et al., 2013; Hu et al., 2020; Meterez et al., 2023).

It is known that such deep skipless networks train slower than their residual counterparts (Martens et al., 2021). Moreover, it is also known that Pre-LN skips downweight residual branches (De & Smith, 2020), which is equivalent to reduced learning rates & downscaled parameter updates from initialisation in linear layers (e.g. Ding et al. (2023); we outline and empirically verify this duality in App. A). This motivates us to study a reparameterisation of the value/projection weights $\mathbf{W}^V, \mathbf{W}^P$:

$$\mathbf{W}^V = \alpha_V\,\mathbf{W}_{\text{init}}^V + \beta_V\,\Delta\mathbf{W}^V, \text{ and } \mathbf{W}^P = \alpha_P\,\mathbf{W}_{\text{init}}^P + \beta_P\,\Delta\mathbf{W}^P, \tag{6}$$

with "skip" $\mathbf{W}_{\text{init}}^V$ fixed to be random orthogonal to preserve the signal propagation achieved at initialisation, and "residual" $\Delta\mathbf{W}^V$ trainable and initialised to zero. We consider downweighting the residuals with fixed $\beta_V \leq \alpha_V = 1$, which biases the matrices $\mathbf{W}^V, \mathbf{W}^P$ to stay closer to initialisation, and would expect $\beta_V = O(\frac{1}{\sqrt{L}})$ to recover the benefits of skip connections (Hayou et al., 2021).[5] . Similar considerations apply for $\mathbf{W}_{\text{init}}^P, \Delta\mathbf{W}^P, \alpha_P, \beta_P$.

In Fig. 3, we find as expected that using smaller $\beta_V$ and $\beta_P$ with this reparameterisation, Eq. (6), already restores much of the training speed loss in skipless attention-blocks, using orthogonally initialised $\mathbf{W}_{\text{init}}^V, \mathbf{W}_{\text{init}}^P$. To close this gap further, we note that from a signal propagation perspective, initialising $\mathbf{W}_{\text{init}}^V, \mathbf{W}_{\text{init}}^P$ to be the identity matrix is equivalent to orthogonal initialisation when the

---

[3]For example, when there is no masking, $C$ becomes the uniform $T \times T$ stochastic matrix: $\frac{1}{T}\mathbb{1}\mathbb{1}^\top$

[4]We set aside the MLP sub-block here for simplicity, but point out that all of our experiments use MLPs so our findings carry over to the full setting.

[5]Although the initial forward pass is identical regardless of $\beta_V$, due to zero initialised $\Delta\mathbf{W}^V$.

attention sub-block is skipless. With identity initialisation $\mathbf{W}_{\text{init}}^V = \mathbf{W}_{\text{init}}^P = \mathbf{I}_d$ we see a consistent improvement over orthogonal initialisation, which essentially matches the Pre-LN block. One thing we can conclude from this experiment, is that restricting the updates to the values and projections from their initialisation replicates the effects of the attention sub-block skip connection, and recovers the lost per-update training speed. We investigate the difference in Fig. 3 of performance between identity and random orthogonal in the appendix (Fig. 15).

## 4.2 REMOVING VALUE AND PROJECTION PARAMETERS

In fact, we can also conclude from Fig. 3 that it is possible to completely remove the value and projection parameters $\mathbf{W}^V, \mathbf{W}^P$ with minimal loss of per-update training speed. Namely, when $\beta_V = \beta_P = 0$ and identity-initialised $\mathbf{W}_{\text{init}}^V = \mathbf{W}_{\text{init}}^P = \mathbf{I}$, we essentially match the Pre-LN block performance after equal numbers of training steps. In this case, we have $\mathbf{W}^V = \mathbf{W}^P = \mathbf{I}$ throughout training, i.e. the values and projection parameters are identity.

To further verify this surprising observation, we consider reparameterised $\mathbf{W}^V, \mathbf{W}^P$, as in Eq. (6) with identity $\mathbf{W}_{\text{init}}^V, \mathbf{W}_{\text{init}}^P$, but now trainable scalars $\alpha_V, \beta_V, \alpha_P, \beta_P$. From an initialisation of $\alpha_V = \alpha_P = 1$ and $\beta_V = \beta_P = 0.2$, we plot the evolution of "residual-skip" ratios $\frac{\beta_V}{\alpha_V}, \frac{\beta_P}{\alpha_P}$ in Fig. 4. Weight decay was not applied on $\alpha_V, \beta_V, \alpha_P, \beta_P$.

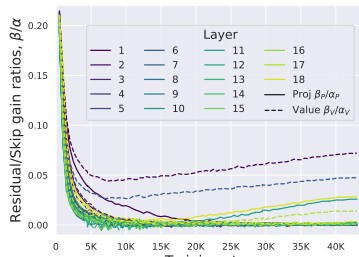

We see that the residual-skip weight ratios $\frac{\beta_V}{\alpha_V}, \frac{\beta_P}{\alpha_P}$ converge to 0 for the vast majority of layers, which indicates that these reparameterised matrices $\mathbf{W}^V, \mathbf{W}^P$ converge to the identity during training. As a result, the extra capacity to perform linear projections via $\mathbf{W}^V, \mathbf{W}^P$ is not

Figure 4: Residual-skip gain ratios $\frac{\beta_V}{\alpha_V}, \frac{\beta_P}{\alpha_P}$ converge to 0 during training.

used. We plot the corresponding trajectories for other scalar parameters like $\beta_{\text{FF}}$, in Figs. 17 to 20, which do not tend to 0. The model in Fig. 4 with trainable $\mathbf{W}^V, \mathbf{W}^P$ achieved worse final evaluation loss than the model in Fig. 3 with identity $\mathbf{W}^V, \mathbf{W}^P$ (1.194 vs. 1.178). Interestingly, this trend is reversed if the attention skip is re-added (Fig. 23).

We thus elect to remove values and projection parameters $\mathbf{W}^V, \mathbf{W}^P$ in our skipless attention sub-blocks, by setting them to the identity.[6] We refer to the resulting sub-block as the *Simplified Attention Sub-block* (SAS). Our full SAS block is depicted in Fig. 10 and we detail the mathematical computation in Eq. (12). We note that SAS blocks use only half of the parameters as well as half the matrix-multiplications in the attention sub-block: only query and key parameters remain. This results in a 13% reduction in the total number of parameters (146M vs 167M for 18 blocks) in the models we consider in this section.[7]

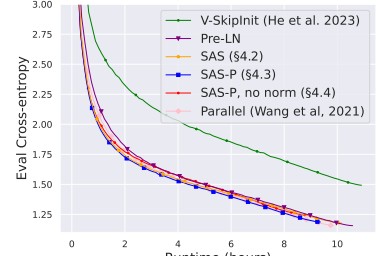

Figure 5: Training speed in terms of runtime. We see our models match (or even slightly outperform) the Pre-LN block.

In Fig. 5 we see that when comparing speed in terms of wall-clock runtime on an A5000 GPU, our SAS block already trains at speeds (slightly) outperforming the default Pre-LN transformer. The corresponding plot comparing speed in terms of training steps taken is provided in Fig. 26. A more detailed analysis of efficiency gains in our simplified blocks can be found in Sec. 5.

Though we do not have a rigorous proof for why the training dynamics in skipless transformers forgo additional capacity by converging to identity value and projection parameters (Fig. 4), nor why fixing such matrices to the identity results in no performance degradation and in fact trains faster than having trainable values and projections (Fig. 3), we offer some half-explanations. First, the fact that $\mathbf{W}^V, \mathbf{W}^P$ are simply linear projections of the input sequence representations $\mathbf{X}$ (as opposed to in the MLP sub-block where elementwise non-linearities are places between such matrices), could

---

[6]The only exception is the first layer's value parameters $\mathbf{W}_1^V$, which is the only ratio above 0.05 in Fig. 4. We saw very minor performance gains by keeping $\mathbf{W}_1^V$ (c.f. Fig. 24), so keep it whilst removing all other $\mathbf{W}_l^V, \mathbf{W}_l^P$ for $l \leq L$

[7]In general, the % of all parameters removed by $\mathbf{W}^V, \mathbf{W}^P$ depends on the ratio of width $d$ to the MLP hidden width $d_{\text{FF}}$, vocabulary size and depth. Here, we have width $d = 768$, MLP hidden width $d_{\text{FF}} = 3072 = 4d$, vocabulary size $50K$ and depth $L = 18$. In the large depth limit, only the ratio $d_{\text{FF}}/d$ matters.

mean that the additional capacity afforded by such matrices is not particularly substantial.[8] This is corroborated by Trockman & Kolter (2023) who found in trained transformers the product $\mathbf{W}^V \mathbf{W}^P$ often has a dominant identity component. Also, from a signal propagation perspective, there is no reason why initialising such matrices to be non-identity (e.g. orthogonal or Gaussian) would be preferred to identity initialisation, nor is it clear why they would be necessary in the first place, especially given the additional matrix-multiplication FLOPs they require.

### 4.3 REMOVING THE MLP SUB-BLOCK SKIP CONNECTION

So far we have simplified the Pre-LN transformer block by removing, without loss of training speed, three key components: 1) the attention sub-block skip connection, as well as 2) value and 3) projection matrices. We next turn to removing the remaining skip connection in the MLP sub-block.

This proved more challenging. Like previous works (Martens et al., 2021; Zhang et al., 2022; He et al., 2023), we found that making activations more linear, motivated through signal propagation, still resulted in a significant loss of per-update training speed without MLP skips when using Adam, as shown in Fig. 25. We also experimented with variants of the Looks Linear initialisation (Balduzzi et al., 2017), with Gaussian, orthogonal or identity weights, to no avail. As such, we use standard activations (e.g. ReLU in this section) and initialisations in the MLP sub-block throughout our work.

Instead, we turn to the idea of parallel MHA and MLP sub-blocks (Zhao et al., 2019; Wang & Komatsuzaki, 2021), which has proven popular in several recent large transformer models, such as PALM (Chowdhery et al., 2022) and ViT-22B (Dehghani et al., 2023). The parallel transformer block is depicted in Fig. 1 (bottom right), and mathematically, given input $\mathbf{X}_{\text{in}}$ it outputs $\mathbf{X}_{\text{out}}$, where:

$$\mathbf{X}_{\text{out}} = \alpha_{\text{comb}} \, \mathbf{X}_{\text{in}} + \beta_{\text{FF}} \, \text{MLP}(\text{Norm}(\mathbf{X}_{\text{in}})) + \beta_{\text{SA}} \, \text{MHA}(\text{Norm}(\mathbf{X}_{\text{in}})), \tag{7}$$

with skip gain $\alpha_{\text{comb}} = 1$, and residual gains $\beta_{\text{FF}} = \beta_{\text{SA}} = 1$ as default.

In the parallel block, the MLP and MHA sub-blocks each take the same (normalised) input, affording more parallelisation compared to the standard Pre-LN block, which computes sub-blocks sequentially. The two sub-blocks are combined by summing their outputs, in conjunction with a single skip connection, with weight $\alpha_{\text{comb}}$. This parallelisation, as well as the removal of one skip connection and one normalisation layer enables efficiency gains: Chowdhery et al. (2022) report the parallel block has 15% faster training speed compared to the standard "sequential" Pre-LN block.

It is straightforward to combine our simplifications from Secs. 4.1 and 4.2 with the parallel block in Eq. (7): we simply 1) use our SAS attention sub-block, Eq. (12), 2) set fixed $\alpha_{\text{comb}} = 0$ to remove all skip connections in the block, and 3) downweight $\beta_{\text{FF}} < 1$. The resulting block is pictured in Fig. 11, and we refer to it as *SAS-Parallel* (SAS-P for short). We see in Fig. 5 that SAS-P trains even faster in terms of runtime compared to the SAS and Pre-LN blocks, and matches the training speed of the parallel block despite using 13% fewer parameters. Our intuition is that the combination of Shaped Attention and identity values/projections preserves signal between blocks throughout training and replaces the need for a skip connection in either sub-block. Moreover, we note that our attention sub-block is the identity function, $\mathbf{X}_{\text{out}} = \mathbf{X}_{\text{in}}$, at initialisation, so there is no difference between our sequential SAS (Fig. 10) and parallel SAS-P (Fig. 11) blocks at initialisation.

### 4.4 REMOVING NORMALISATION LAYERS

The final simplification we consider is removing normalisation layers, leaving us with our simplest block (Fig. 1, top right). From a signal propagation initialisation perspective, normalisation has been expendable at all stages of our simplifications in this section: the idea is that normalisation in Pre-LN blocks implicitly downweights residual branches, and this beneficial effect can be replicated without normalisation by another mechanism: either explicitly downweighting residual branches when skips are used, or biasing attention matrices to the identity/transforming MLP non-linearities to be "more" linear otherwise. As we account for these mechanisms in our modifications (downweighted MLP $\beta_{\text{FF}}$ & Shaped Attention), from an initialisation perspective there is no need for normalisation.

Of course, these modifications have effects on training speeds and stability beyond initialisation, which are harder to predict from existing theory alone. In Fig. 5 we see that removing normalisa-

---

[8]E.g., in single-head attention one can reparameterise $\mathbf{W}^V, \mathbf{W}^P$ into one matrix with no expressivity loss.

tion allows even our simplest transformer block, which does not have skips, sequential sub-blocks, values, projections or normalisations, to match the training speed of the Pre-LN block in terms of runtime. Having said that, we do observe a slight degradation in training speed per iteration, as seen in Fig. 26, suggesting that normalisation layers have some beneficial properties for training speed beyond what is captured by signal propagation theory. We thus treat our SAS (Fig. 10) and SAS-P (Fig. 11) blocks, with normalisation, as our main approaches. On this note, we point out that Dehghani et al. (2023) found extra normalisation on the queries and keys to provide improved training stability in ViT-22B, going against the recent trend of researchers seeking to remove normalisation.

## 5 FURTHER EXPERIMENTAL ANALYSIS

Having introduced all of our simplifications in Sec. 4, we now provide further empirical analysis of our simplified blocks across a range of settings, as well as details of the efficiency gains afforded by our simplifications. In interest of space, additional experimental details can be found in App. D.

**Depth Scaling** Given that signal propagation theory often focuses on large depths, where signal degeneracies usually appear, it is natural to ask whether the improved training speeds of our simplified transformer blocks also extend to larger depths. In Fig. 6, we see that scaling depth from 18 to 72 blocks leads to an increase in performance in our models as well as the Pre-LN transformer, indicating that our simplified models are able to not only train faster but also to utilise the extra capacity that more depth provides. Indeed, the per-update trajectories of our simplified blocks and Pre-LN are near-indistinguishable across depths, when using normalisation.

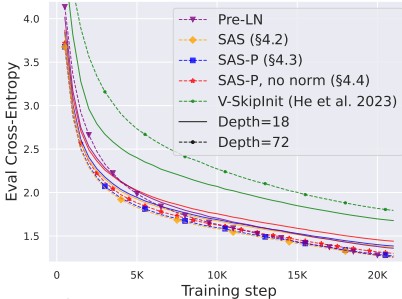

Figure 6: Our models improve when deeper (dashed, marked lines) vs. shallower (solid lines), unlike V-SkipInit (He et al., 2023).

On the other hand, we see that Value-SkipInit (He et al., 2023) actually trains slower per update at depth 72 compared to 18 despite the increase in capacity and parameter count. Moreover, the gap in performance between Value-SkipInit and the other models increases with larger depth, which implies poor scalability of the previous method. We note that 72 blocks is already reasonably deep by publically-available modern standards (Hoffmann et al., 2022; Touvron et al., 2023).

**BERT** Next, we demonstrate our simplified blocks' performance extends to different datasets and architectures besides autoregressive decoder-only, as well as on downstream tasks. We choose the popular setting of the bidirectional encoder-only BERT model Devlin et al. (2018) for masked language modelling, with downstream GLUE benchmark.

In particular, we adopt the "Crammed" BERT setup of Geiping & Goldstein (2023), which asks how well one can train a BERT model with a modest training budget: 24 hours on a single consumer GPU. The authors provide an architecture, data pipeline and training setup that has been optimised for this low resource setting. We note that the Crammed architecture uses the Pre-LN block, and describe other setup details in App. D. We plug-in our simplified blocks, keeping the existing optimised hyperparameters, besides tuning learning rate and weight decay.

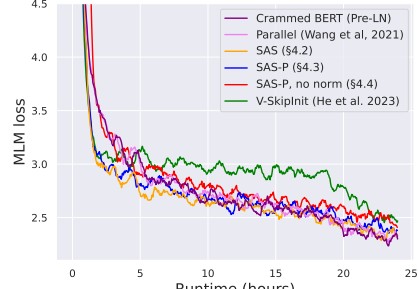

Figure 7: Masked language modelling loss vs runtime on a 2080Ti GPU for 24 hours.

In Fig. 7, we see that our simplified blocks (especially with normalisation) match the pre-training speed on the masked language modelling task compared to the (Crammed) Pre-LN baseline within the 24 hour runtime. On the other hand, the removal of skip connections without modifying the values and projections (as in He et al. (2023)) once again leads to a significant loss of training speed. In Fig. 27, we provide the equivalent plot in terms of microbatch steps.

Moreover in Table 1, we find that our methods match the performance of the Crammed BERT baseline after finetuning on the GLUE benchmark. We provide a breakdown over the downstream tasks in Table 2. We use the same finetuning protocol as Geiping & Goldstein (2023) (5 epochs, constant hyperparameters across tasks, dropout regularisation) for a fair comparison. Interestingly,

Value-SkipInit is largely able to recover from its poor pre-training in the fine-tuning phase. This, combined with the need for dropout when fine-tuning, suggests that factors besides pre-training speed are also important for fine-tuning. As the focus of our work primarily concerns training speed from random initialisations, we leave this to future work. Relatedly, we found removing normalisations (Sec. 4.4) to cause instabilities when fine-tuning, where a small minority of sequences in some downstream datasets had NaN values in the initial forward pass from the pre-trained checkpoint.

**Efficiency Gains**   In Table 1, we also detail the parameter count and training speeds of models using different Transformers blocks on the masked language modelling task. We compute the speed as the ratio of the number of microbatch steps taken within the 24 hours of pre-training, relative to the baseline Pre-LN Crammed BERT. We see that our models use 16% fewer parameters, and SAS-P & SAS are 16% & 9% faster per iteration, respectively, compared to the Pre-LN block in our setting. We note that in our implementation the Parallel block is only 5% faster than the Pre-LN block, whereas

Table 1: GLUE benchmark & efficiency gains. Our SAS & SAS-P match the downstream performance of the Pre-LN baseline up to statistical significance over 3 seeds, but use 16% fewer parameters and enjoy up to 16% faster throughput.

| Block | GLUE | Params | Speed |
|---|---|---|---|
| Pre-LN (Crammed) | $78.9_{\pm.7}$ | 120M | 1 |
| Parallel | $78.5_{\pm.6}$ | 120M | 1.05 |
| V-SkipInit | $78.0_{\pm.3}$ | 120M | 0.95 |
| SAS (Sec. 4.2) | $78.4_{\pm.8}$ | 101M | 1.09 |
| SAS-P (Sec. 4.3) | $78.3_{\pm.4}$ | 101M | 1.16 |
| SAS-P, no norm | - | 101M | 1.20 |

Chowdhery et al. (2022) observed 15% faster training speeds, suggesting that further throughout increases may be possible with a more optimised implementation. Our implementation, like Geiping & Goldstein (2023), uses automated operator fusion in PyTorch (Sarofeen et al., 2022)

**Longer training**   Finally, given the current trends of training smaller models for longer on more data (Hoffmann et al., 2022; Touvron et al., 2023), we investigate if our simplified blocks continue to match the training speeds of the Pre-LN block with longer training. To do this, we take our models from Fig. 5 on CodeParrot and train with $3\times$ tokens. To be precise, we train for around 120K (rather than 40K) steps with batch size 128 and sequence length 128, which results in around 2B tokens. In Fig. 8, we do indeed see that our simplified SAS and SAS-P blocks continue to match or outperform the Pre-LN block in training speed when trained on more tokens.

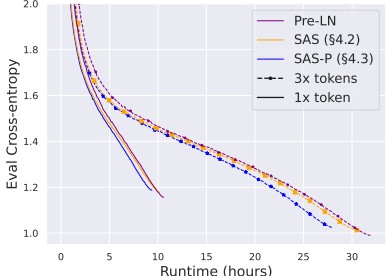

Figure 8: Training speeds continue to hold with longer training.

## 6   DISCUSSION

**Limitations and future work**   While we have demonstrated the efficacy of our simplifications across architectures, datasets, and tasks, the models we have considered (100-300M parameters) are small relative to the largest transformers. It would be interesting to investigate the performance of our simplified blocks at larger scales, especially because Chowdhery et al. (2022) report parallel blocks improve relative to Pre-LN blocks with scale. Our depth scaling experiments already show promise in this regard. On the theoretical side, though we were able to match the training speed of Pre-LN blocks with normalisation removed (Fig. 5), there are still unanswered questions regarding the benefits of normalisation for training speed and stability, and we were unable to remove normalisation with good downstream task performance. Moreover, while we tuned key hyperparameters like learning rate, it is possible that many default hyperparameters and choices we inherited, e.g. the AdamW optimiser, or fine-tuning protocol, are overfit to the default Pre-LN block, and an exhaustive hyperparameter search for our simplified blocks would yield further improvements. Finally, on the practical side, we believe that a more hardware-specific implementation of our simplified blocks could give further improvements to training speed and performance.

**Conclusion**   In this work, we asked whether it is possible to simplify the standard Transformer block by removing unnecessary components. Combining signal propagation theory and empirical insights, we have shown that it is possible to remove skip connections, sequential sub-blocks, value and projection parameters, without loss of training speed or downstream task performance. As a result, our models have around 15% fewer parameters and 16% increased throughput. We believe our work can lead to simpler architectures being used in practice, thereby helping to bridge the gap between theory and practice in deep learning, and reducing the cost of large transformer models.

REPRODUCIBILITY STATEMENT

Our code for experiments on auto-regressive transformers can be found at `https://github.com/bobby-he/simplified_transformers`.

ACKNOWLEDGMENTS

We would like to thank Sotiris Anagnostidis, Andrei Ivanov & Lorenzo Noci for helpful discussions in the initial stages of this project, and James Martens, John Martinis, Keivan Mohtashami, Tiago Pimentel & Imanol Schlag, as well as the anonymous reviewers, for constructive feedback on an early version of this manuscript.

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

## A  DUALITY BETWEEN DOWNWEIGHTED RESIDUALS AND RESTRICTING UPDATES IN LINEAR LAYERS

In Sec. 4.1, we motivated our reparameterisation of the value and projection parameters, Eq. (6), through a duality between downweighted residuals branches and restricting parameter updates (materialised through smaller learning rates) in linear layers. This is a relatively simple argument, found elsewhere in the literature e.g. Ding et al. (2023), which we outline here for completeness.

We suppose we have a (differentiable) loss function $L(W)$, which is a function of some parameter matrix $W$. We consider taking a gradient step to minimise $L$, with learning rate $\eta_W$ from initialisation $W_0$. This would give new parameters $W_1$:

$$W_1 = W_0 - \eta_W \left. \frac{dL}{dW} \right|_{W=W_0} \tag{8}$$

Now suppose we have a reparameterisation of the parameters to $W'$, with the same loss $L(W')$ as before:

$$W' = U + \beta V \tag{9}$$

for fixed scalar $\beta$, fixed matrix $U$ and trainable parameter matrix $V$. We let $V$ be initialised to $V_0$, satisfying $W_0 = U + \beta V_0$.

If we take a gradient step in $V$ with learning rate $\eta_V$, then we get new parameters $V_1$:

$$
\begin{aligned}
V_1 &= V_0 - \eta_V \left. \frac{dL}{dV} \right|_{V=V_0} \\
&= V_0 - \eta_V \cdot \left. \frac{dW'}{dV} \right|_{V=V_0} \cdot \left. \frac{dL}{dW'} \right|_{W'=W_0} \\
&= V_0 - \eta_V \beta \left. \frac{dL}{dW'} \right|_{W'=W_0} \\
&= V_0 - \eta_V \beta \left. \frac{dL}{dW} \right|_{W=W_0}
\end{aligned}
\tag{10}
$$

where in the last line we just relabelled the reparameterisation variable $W'$ to $W$.

Feeding Eq. (10) back into Eq. (9), we obtain:

$$
\begin{aligned}
W_1' &= U + \beta V_1 \\
&= U + \beta V_0 - \eta_V \beta^2 \left. \frac{dL}{dW} \right|_{W=W_0} \\
&= W_0 - \eta_V \beta^2 \left. \frac{dL}{dW} \right|_{W=W_0}
\end{aligned}
\tag{11}
$$

due to the equivalence of initialisations.

To match $W_1'$, Eq. (11), with $W_1$, Eq. (8), we require:

$$\eta_W = \eta_V \beta^2.$$

Thus, any gradient step we take in the reparameterisation, $W' = U + \beta V$, corresponds to taking the same gradient step in original parameterisation, $W$, but with a learning rate scaled by $\beta^2$. If $\beta < 1$, as is the case in Pre-LN residual branches, this corresponds to downscaling the learning rate.

In the context of our reparameterisation of values and projection parameters Eq. (6), this is then equivalent to downscaling the learning rates of $\mathbf{W}^V, \mathbf{W}^P$ by $\beta_V^2, \beta_P^2$, if using (stochastic) gradient descent. With AdamW (Loshchilov & Hutter, 2017), one factor of $\beta$ gets divided out by the preconditioner, so the reparameterisation acts as if we scale the learning rate by $\beta$ not $\beta^2$.

To verify this theoretical duality empirically, we plot the equivalent of Fig. 3 but where, instead of reparameterisation (Eq. (6)) with varied $\beta$, we reduce the learning rate of the value and projection parameters (keeping the learning rate of other parameters fixed). As expected, we see that reducing the ratio of learning rate for value/projection parameters compared to other parameters improves the training speed, just like the downweighted residual reparametersiation (Fig. 3).

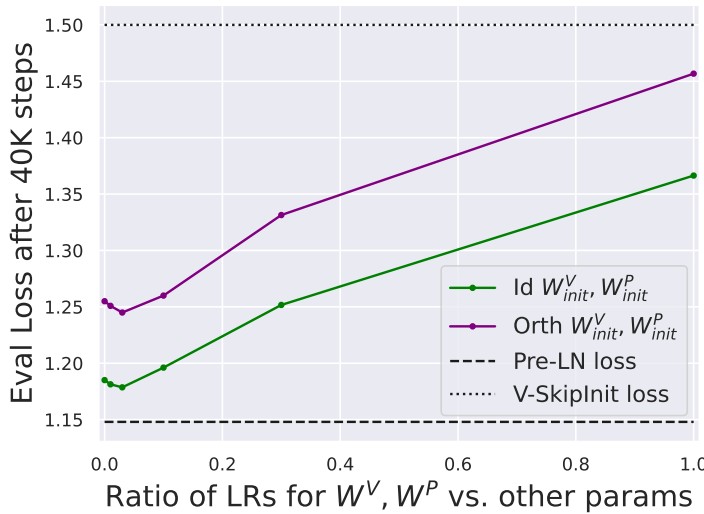

Figure 9: Equivalent of Fig. 3 that empirically confirms the duality between downweighted residuals and reduced learning rates.

## B    BLOCK LAYOUTS

In Fig. 10 and Fig. 11 we show the layouts of our SAS block (Sec. 4.2) and parallel SAS-P block (Sec. 4.3). These are the equivalent plots to the layouts in Fig. 1.

Mathematically, our SAS attention sub-block computes (in the notation of Eq. (2)):

$$\mathbf{X}_{\text{out}} = \widetilde{\text{MHA}}(\text{Norm}_1(\mathbf{X}_{\text{in}})), \quad \text{where} \quad \widetilde{\text{MHA}}(\mathbf{X}) = \text{Concat}\big(\widetilde{\text{Attn}}_1(\mathbf{X}), \dots, \widetilde{\text{Attn}}_H(\mathbf{X})\big),$$

(12)

$$\widetilde{\text{Attn}}_h(\mathbf{X}) = (\alpha_h \mathbf{I}_T + \beta_h \mathbf{A}_h(\mathbf{X}) - \gamma_h C)\mathbf{X}_h, \quad \text{and} \quad \mathbf{A}_h(\mathbf{X}) = \text{SM}\left(\frac{1}{\sqrt{d_k}}\mathbf{X}\mathbf{W}_h^Q \mathbf{W}_h^{K\top}\mathbf{X}^\top + \mathbf{M}\right).$$

(13)

Here, $\mathbf{X}_h \in \mathbb{R}^{T \times \frac{d}{H}}$ are column blocks of $\mathbf{X} \in \mathbb{R}^{T \times d}$, i.e. $\mathbf{X} = \text{Concat}(\mathbf{X}_1, \dots, \mathbf{X}_H)$, & SM is Softmax.

## C    ADDITIONAL EXPERIMENTS

In this section, we provide additional experiments and ablations on top of those provided in the main paper. The experiments in this section are ordered to follow the chronological order of where they are referenced (or most relevant) in the main paper.

**Linear vs Cosine decay LR schedule**    Fig. 12 compares linear and cosine decay LR schedule. We see that linear decay provides better final performance across both our models and baselines, and use linear decay throughout the rest of the paper.

**Shaped Attention vs Value-SkipInit**    Fig. 13 explains our reasons for using Shaped Attention Eq. (5) (Noci et al., 2023) over the modified attention matrix, $\alpha \mathbf{I} + \beta \mathbf{A}(\mathbf{X})$, Eq. (4), that was introduced by He et al. (2023) in Value-SkipInit. We see that Shaped Attention gives a small but

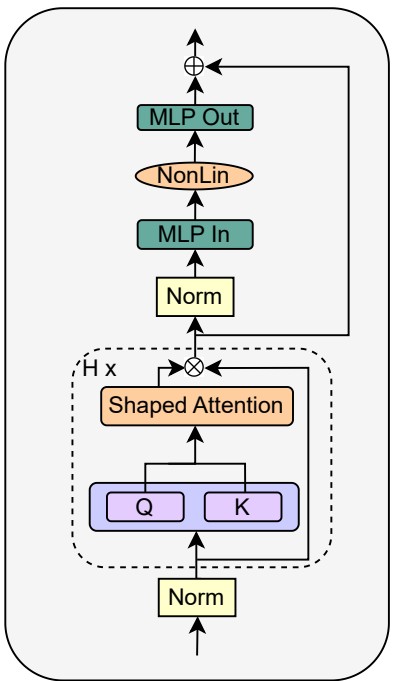

Figure 10: The SAS block that we obtain at the end of Sec. 4.2.

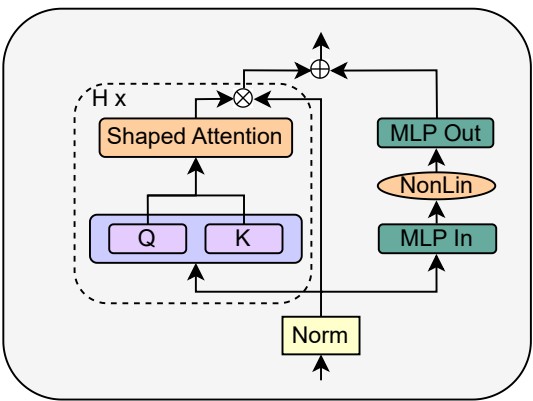

Figure 11: The SAS-P block, with normalisation, that we obtain at the end of Sec. 4.3.

consistent gain throughout training. The experiments here follow the same training and hyperparameter setup as those in Sec. 4.

**Sensitivity to MLP block gain initialisation** In Sec. 4.1, we motivated downweighting the initialisation of trainable MLP block weight $\beta_{\text{FF}}$ (c.f. Eqs. (1) and (7)) in skipless architectures to replicate the implicit downweighting mechanism of Pre-LN skips. Fig. 14 shows the sensitivity of final loss to our initialisation for trainable $\beta_{\text{FF}}$.

**Figure 3 with tied orthogonals** In Fig. 3, we observed that restricting the updates to value and projection parameters recovers nearly all of the lost training speed in Transformer blocks without attention sub-block skips. This phenomenon occurred for both random orthogonal and identity initialisations $\mathbf{W}_{\text{init}}^V, \mathbf{W}_{\text{init}}^P$, but identity initialisation outperformed orthogonal, which may be a bit surprising as they should be identical from a signal propagation perspective.

To investigate this, we consider two alternatives:

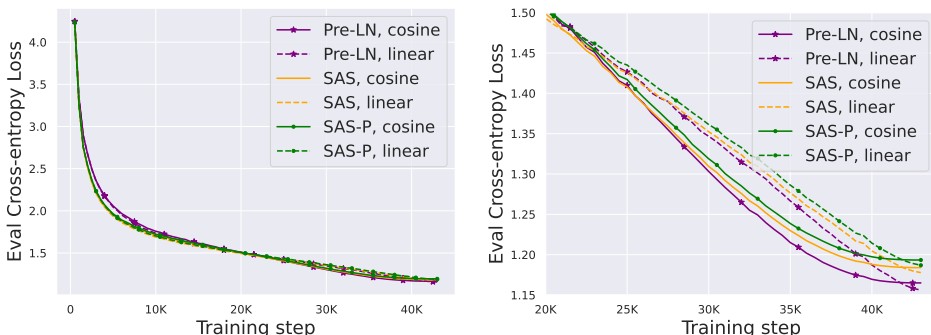

Figure 12: Comparing training performance with cosine and linear decay LR schedulers on Code-Parrot. The right plot is a zoomed-in version of the left. We see that linear decay consistently provides a better final performance than cosine decay, despite trailing for most of the steps towards the end of training.

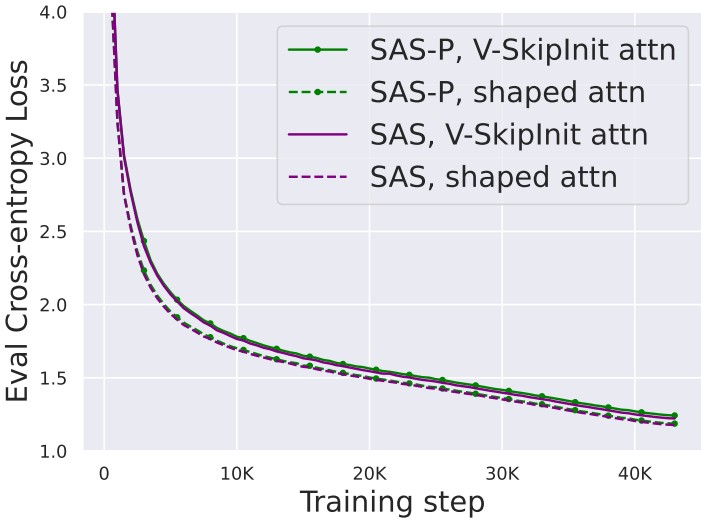

Figure 13: Shaped Attention (dashed lines) provides a small performance boost compared to the attention matrix of Value-SkipInit (solid lines), for both SAS and SAS-P blocks. All transformers are 18-Layer autoregressive GPT models, and the dataset is CodeParrot.

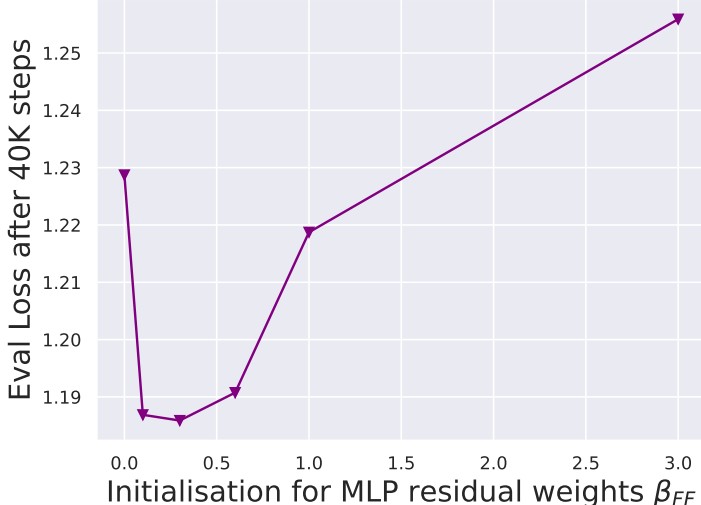

Figure 14: Final test loss achieved as a function of the initialisation for trainable MLP block gains $\beta_{\text{FF}}$ on CodeParrot.

1. **"LL-tied"** or last-layer tied: this is when we initialise all but the final layer projection matrix as independent random orthogonals. Then for the final layer projection matrix $\mathbf{W}_{\text{init},L}^{P}$ (or rather its transpose), we tie the initialisation to the previous layers, as:

$$\mathbf{W}_{\text{init},L}^{P\top} = \left( \prod_{l=1}^{L-1} \mathbf{W}_{\text{init},l}^{V} \mathbf{W}_{\text{init},l}^{P} \right) \mathbf{W}_{\text{init},L}^{V}.$$

The purpose of this is so that the combined product over all layers $\prod_{l=1}^{L} \mathbf{W}_{\text{init},l}^{V} \mathbf{W}_{\text{init},l}^{P}$ is the identity, which mimics the functional output of the whole transformer with identity $\mathbf{W}_{\text{init},l}^{V} = \mathbf{I} = \mathbf{W}_{\text{init},l}^{P}$ (up to the MLP blocks, but we note that the MLP weights are independently Gaussian initialised and so are rotationally invariant at initialisation, and are also downweighted as they lie on a downweighted residual branch).

2. **"All-tied"**: this is when for each attention layer/sub-block $l$ we have $\mathbf{W}_{\text{init},l}^{V} = \mathbf{W}_{\text{init},l}^{P\top}$ with random orthogonal initialisation, which makes the value-projection product identity, $\mathbf{W}_{\text{init},l}^{V} \mathbf{W}_{\text{init},l}^{P} = \mathbf{I} \ \forall l \leq L$, and hence matches the outputs of each attention sub-block exactly as if we had identity values and projections at initialisation. This initialisation is similar to that of Trockman & Kolter (2023), although here we are considering skipless attention sub-blocks.

Fig. 15 is the equivalent of Fig. 3 but with LL-tied (yellow line with diamond markers) and all-tied (blue line with star markers) included. In Fig. 15, we see that matching the functional output (as in LL tied) with orthogonal initialisation provides a slight improvement to close the gap between the random orthogonal (green line) and identity (purple line) initialisations. Matching the attention sub-block outputs (as in all-tied) further improves performance, but does not fully close the gap to identity initialisation. Interestingly, it seems like orthogonally initialised values and projections do benefit from being trainable (i.e. small but non-zero $\beta_V, \beta_P$). We leave a further exploration of these observations to future work.

**Further scalar parameter trajectories** In Fig. 4 we saw that residual-skip gain ratios on the values and projections $\frac{\beta_V}{\alpha_V}, \frac{\beta_P}{\alpha_P}$ (from the reparameterisation in Eq. (6)) converge to zero during training for the vast majority of layers, in models without attention sub-block skip connections. In Fig. 16 we plot the corresponding plot to Fig. 4 but for a parallel block with no skip connections (i.e. SAS-P with reparameterisation Eq. (6) and trainable values and projections. Like before, we also initialise trainable $\beta_V, \beta_P$ to 0.2, and $\alpha_V, \alpha_P$ to 1). Again, we see that the vast majority of

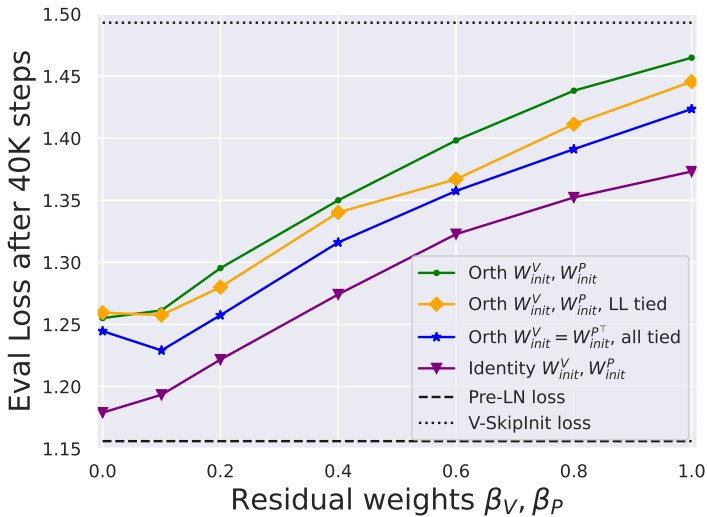

Figure 15: Equivalent of Fig. 3 but with tied orthogonal initialisations.

$\frac{\beta}{\alpha}$ ratios converge to 0, indicating that the value and projection parameters converge to the identity. Also, again we see that the first value matrix is the only ratio above $0.05$ at the end of training.

Figs. 17 to 20 plot the corresponding trajectories, of the model in Fig. 4, for various other trainable scalar parameters we have: namely, the three shaped attention parameters 1) $\alpha$ on the identity (Fig. 17), 2) $\beta$ on the softmax attention output (Fig. 18), 3) $\gamma$ on the centring matrix (Fig. 19), as well as 4) $\beta_{\text{FF}}$ on the MLP block (Fig. 20). We see that none of these other scalar parameters converge to zero. The shaped attention parameters have error bars denoting standard deviations across (the 12) heads.

In Fig. 21, we plot the equivalent of Fig. 4 but for an attention skipless model with random orthogonally initialiased (from the Haar measure) value and projection weights. Again, we see that the vast majority of residual/skip gain ratios converge from their initial value of 0.2 to 0 during training albeit with slightly more outliers than in Fig. 4.

However, in Fig. 22 we plot the equivalent of Figs. 4 and 21 but for a Pre-LN model that has attention skip connection (and identity initialised values/projections). Now, we see that the introduction of the skip connection encourages many more residual/skip gain ratios in the value/projection weights to increase during training (notice the y-axis), i.e. encouraging the values/projections to leave their identity initialisation. Together, these results reaffirm the complex interactions between skip connections and value/projection weights in the standard transformer architecture highlighted by our work.

**Identity values and projections with default Pre-LN block**  In Sec. 4.2 we observed that removing values and projection parameters by setting them to the identity improves the convergence speed per parameter update of transformer blocks with skipless attention sub-blocks. This raises the question of whether the same would occur in the standard block that uses attention sub-blocks skip connections i.e. the Pre-LN block. In Fig. 23 we compare the default Pre-LN block to one with values and projections set to be identity. We see that in this case identity values and projections actually slightly hurt performance in terms of loss reduction per update, in constrast to the skipless case. We also tried to downweight the attention block residual ($\beta_{\text{SA}} < 1$ in the notation of Eq. (1)) with identity values and projections to see if the scale of the attention skip and residual was the reason for this difference, but this did not change the findings of Fig. 23. We do not have a satisfying explanation for this, but our intuition is that identity value and projections (as opposed to e.g. Gaussian/random orthogonal initialisation) with attention skip means that the two branches of the skip are no longer independent at initialisation and interfere with each other, though it is unclear that this would continue to hold during training.

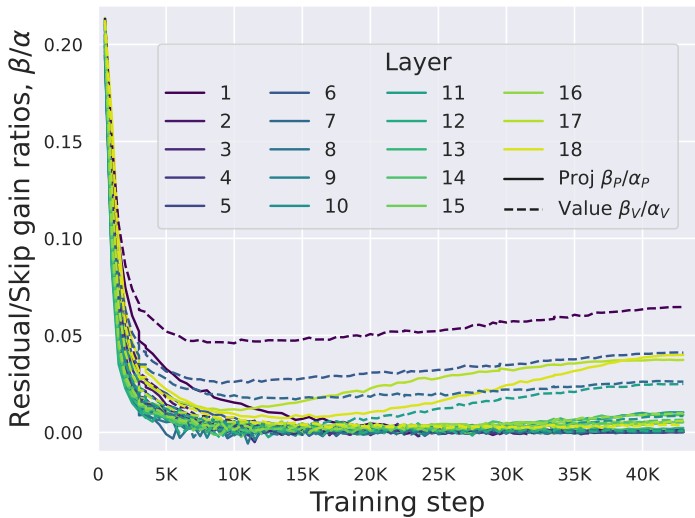

Figure 16: Corresponding plot to Fig. 4 but for a parallel block with no skips.

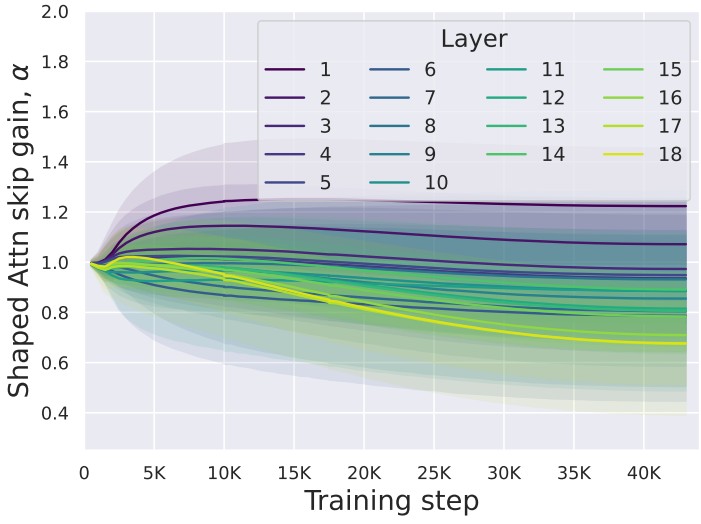

Figure 17: Trajectories for shaped attention $\alpha$ parameter.

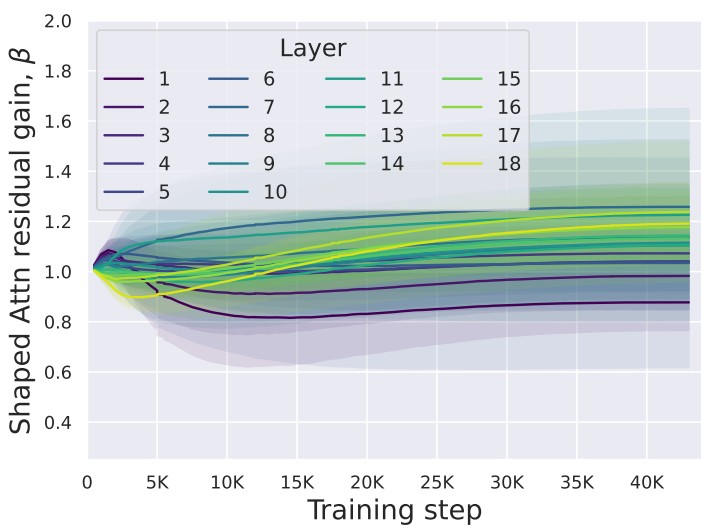

Figure 18: Trajectories for shaped attention $\beta$ parameter.

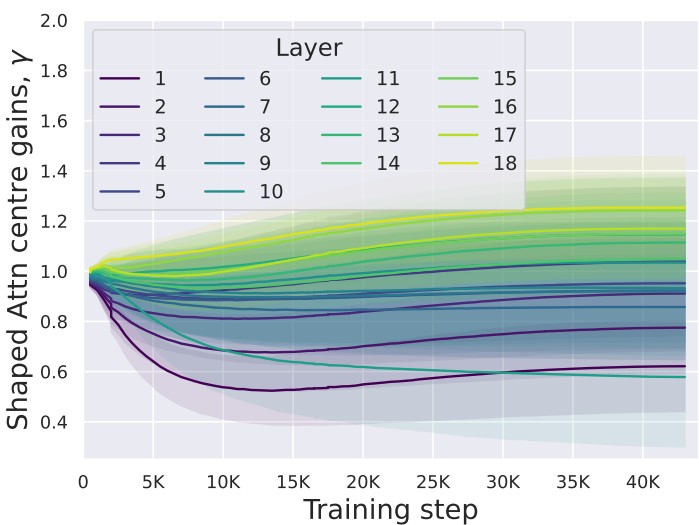

Figure 19: Trajectories for shaped attention $\gamma$ parameter.

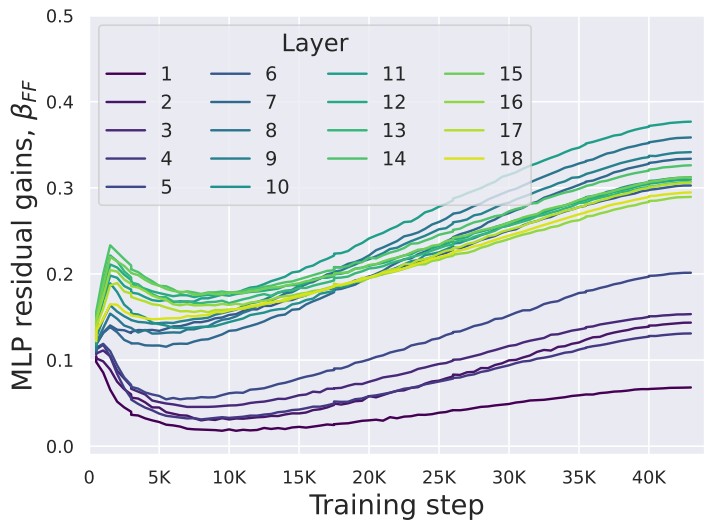

Figure 20: Trajectories for MLP block $\beta_{\text{FF}}$ parameter.

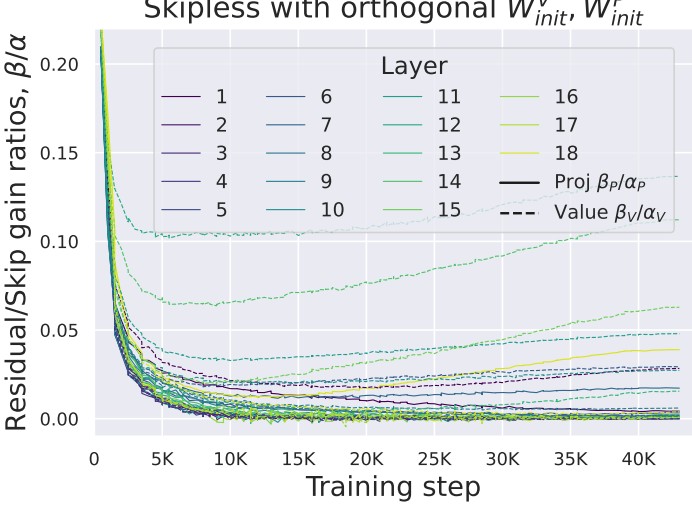

Figure 21: Trajectories for MLP block $\beta_{\text{FF}}$ parameter.

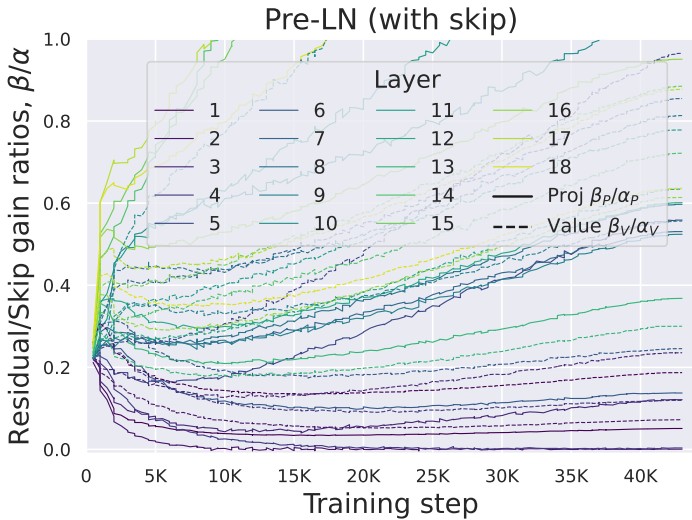

Figure 22: Trajectories for MLP block $\beta_{\mathrm{FF}}$ parameter.

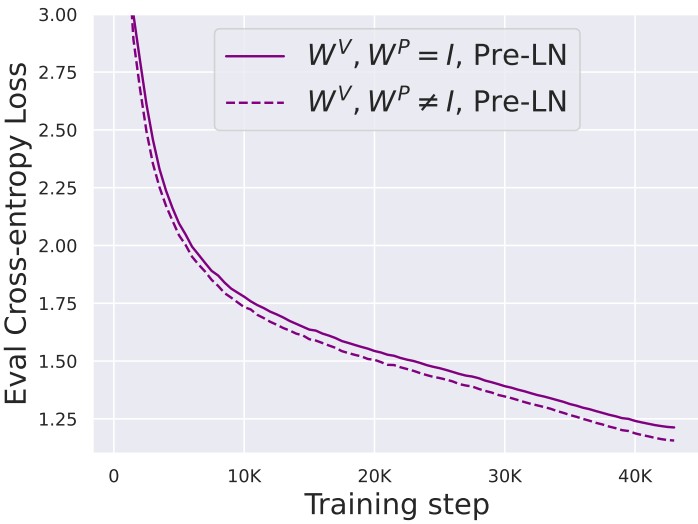

Figure 23: Pre-LN block performs worse when setting values and projections to identity, unlike in the skipless setting.

**Using the first value matrix**    In Figs. 4 and 16 we see that the vast majority of value and projection parameters stay close to the identity during training when initialised to identity, even when they have the capacity to move away from initialisation. The first layer value matrix $\mathbf{W}_1^V$ is an exception to this rule. In Fig. 24 we see that allowing the first layer value parameters to be trainable provides a very small boost to training performance, when all other value and projection weights are fixed to the identity. Intuitively it makes sense that the first layer value parameters would be more important than others because they act directly on the input embedding at the beginning of the model. We thus choose to reincorporate trainable value parameters (using identity initialisation in Eq. (6)) in the first layer of our models using SAS and SAS-P blocks, but remove all other values $\mathbf{W}_l^V$ for $l > 1$, and all projections too $\mathbf{W}_l^P \; \forall l \geq 1$, by fixing to the identity.

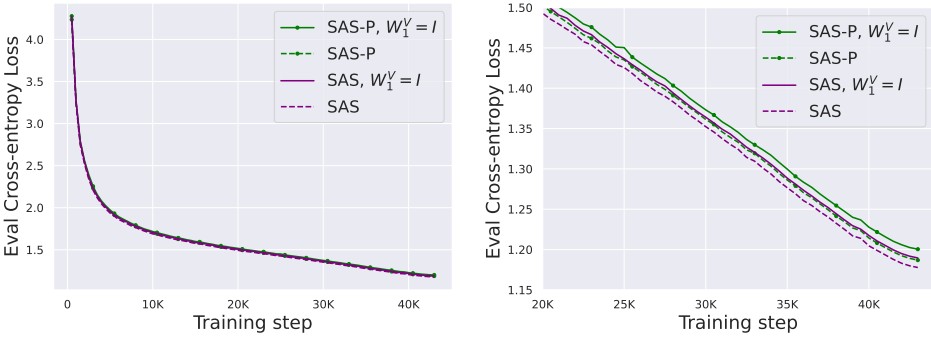

Figure 24: Comparing training performance with trainable first layer values $\mathbf{W}_1^V$ vs with identity first layer values $\mathbf{W}_1^V$, for models using our SAS and SAS-P blocks. All other values and projections are set to the identity. The right plot is a zoomed-in version of the left. We see that having trainable first layer values provides a (very small) boost in performance.

**Linearising MLP activations**    As stated in Sec. 4.3, we tried to use the recent idea of "linearising" activation functions in order to obtain better signal propagation (Martens et al., 2021; Zhang et al., 2022; Li et al., 2022) in deep NNs with skip connections, and recover lost training speed when the MLP skip is removed. In particular, Li et al. (2022) show for Leaky ReLU,

$$\mathrm{LReLU}(x) = \max(x, sx),$$

with negative slope $s \in [0, 1]$, we need $s = 1 - O(\frac{1}{\sqrt{L}})$ to obtain well behaved signal propagation in MLPs at large depths $L$.

In Fig. 25, we took our 18-block model trained with SAS block (Fig. 10), and assessed training performance without the MLP skip, $\alpha_{\mathrm{FF}} = 0$. We tried 3 different activations: 1) standard ReLU, 2) LReLU with slope $s = 0.2$, and 3) LReLU with $s = 0.8 \approx 1 - \frac{1}{\sqrt{18}}$.

We see that all blocks without MLP skip train significantly slower than our SAS block (which matches the training speed of the Pre-LN block). In fact, linearising ReLU into LReLU seemed to hurt training speed rather than help it. These findings are consistent with those of previous works with AdamW optimiser (Martens et al., 2021; Zhang et al., 2022; He et al., 2023). We note that a big reason behind this is that the architectures with skipless MLP sub-blocks required an order of magnitude smaller learning rate ($1e - 4$ vs $1e - 3$) otherwise training was unstable.

**Loss vs training step**    In Fig. 26, we provide the equivalent plot to Fig. 5, but in terms of loss over the steps taken. Our SAS and SAS-P essentially match the Pre-LN model in terms of loss reduction per step, whilst removing normalisation slightly hurts performance.

**Crammed Bert loss vs training step**    In Fig. 27 we plot the MLM loss in the Crammed Bert setting on the Pile dataset, as a function of the number of microbatch steps taken. Because our models have higher throughput they are able to take more steps within the 24 hour allotted time.

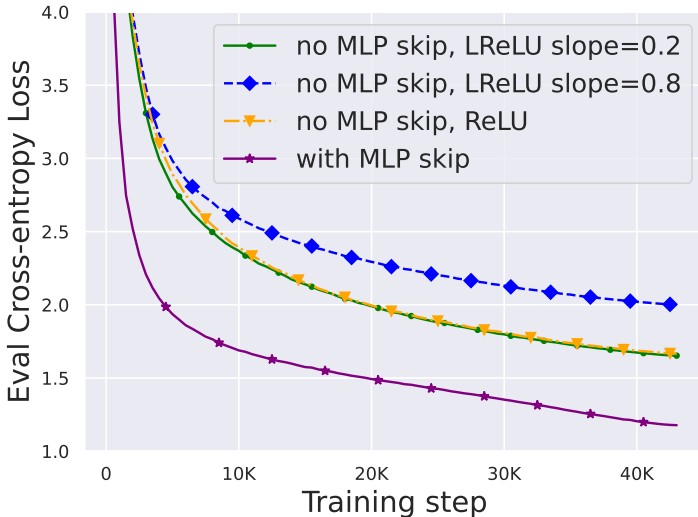

Figure 25: Removing MLP skips results in significant losses of training speed, even when linearising activations.

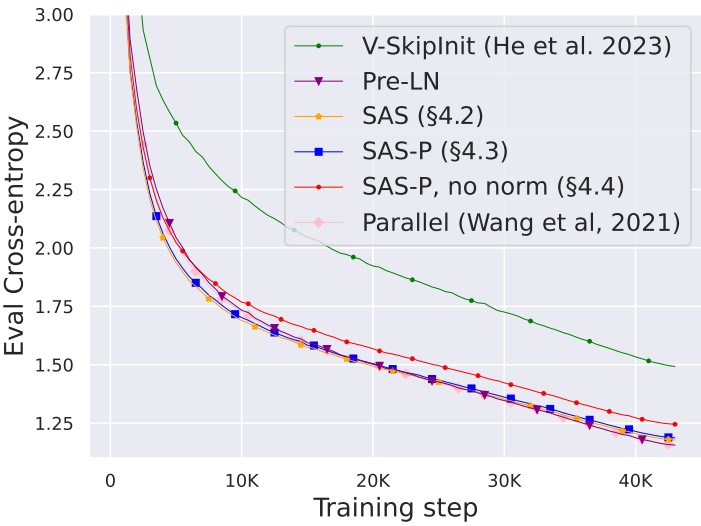

Figure 26: Equivalent of Fig. 5 but with steps on the x-axis.

**GLUE breakdown**    In Table 2, we provide a breakdown of the GLUE results in Table 1 in terms of different tasks.

**Autoregressive Language Modelling**    We investigate if our findings hold in the next-token pre-diction language modelling domain. This also allows us to test at larger sequence lengths (512) than other experiments in this work (128). The task is autoregressive language modelling on the Languini Benchmark books dataset (Stanić et al., 2023), and we use the same codebase and tokeniser provided by the authors. Models have 12 layers with width 768, which gives ~100M parameters (including tied embedding/unembedding) by default when MLP width is 3072 (4×768). Sequence length is 512 and we train for 19K steps on batch size 128, giving 1.2B training tokens. Learning rate is lin-early warmed up for 500 steps to a maximum value that is tuned for all models separately (3e-3 for our simplified models, 1e-3 for the default blocks), before linear decay. ALiBi positional encoding

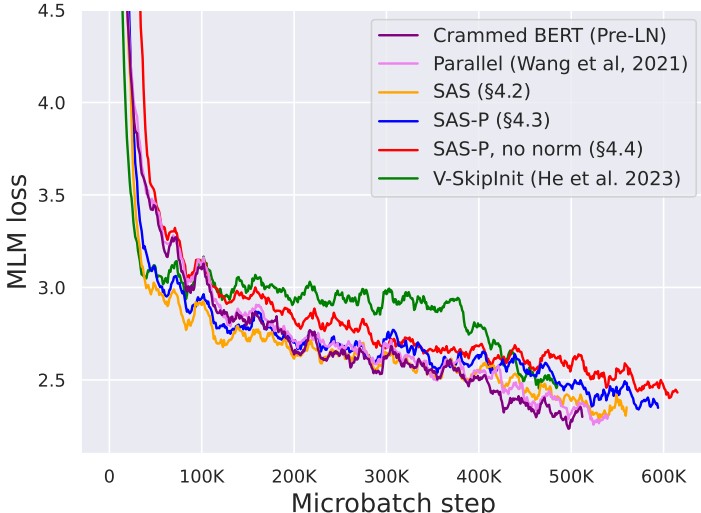

Figure 27: MLM loss vs microbatch steps taken. Note that because the LR schedule depends on the total number of steps taken by a model, and is different in different models for the same number of steps taken, comparing models in terms of MLM loss at a given step is not so informative.

Table 2: Breakdown of GLUE results on different tasks. Results are the mean over 3 seeds.

|  | GLUE | MNLI | SST-2 | STSB | RTE | QNLI | QQP | MRPC | CoLA |
|---|---|---|---|---|---|---|---|---|---|
| Pre-LN (Crammed) | $78.9_{\pm.7}$ | 81.2/81.6 | 89.9 | 87.6 | 56.3 | 88.9 | 87.1 | 88.3 | 49.4 |
| Parallel | $78.5_{\pm.6}$ | 80.8/80.9 | 91.0 | 87.5 | 54.9 | 88.0 | 87.0 | 87.3 | 49.0 |
| SAS (Sec. 4.2) | $78.4_{\pm.8}$ | 79.7/80.1 | 90.3 | 84.7 | 58.4 | 87.5 | 86.8 | 87.5 | 50.6 |
| SAS-P (Sec. 4.3) | $78.3_{\pm.4}$ | 79.5/79.4 | 90.8 | 85.2 | 59.1 | 87.5 | 86.5 | 86.0 | 50.5 |
| V-SkipInit | $78.0_{\pm.3}$ | 79.7/80.4 | 90.4 | 85.1 | 54.4 | 87.0 | 86.2 | 87.9 | 51.5 |

and GeLU activations are used. Training takes place on a single RTX-2080Ti (with microbatches of size 16), and we use AdamW with weight decay 0.1.

We plot a training speed comparison of our simplified blocks against default in Fig. 28 below, in terms of runtime on the x-axis and evaluation perplexity on the y-axis. We again see that our models are able to match the training speed of default Pre-LN and Parallel blocks. Moreover, our SAS block (orange curve with star markers) achieves the same final perplexity after 19K steps (22.37 vs 22.39) as the Parallel block (pink curve with diamond markers) despite using 15% fewer parameters.

## D    IMPLEMENTATION DETAILS

In this section we add remaining implementation details that were not discussed in the main paper. We break down our implementation details into two subsections, one for the next-token prediction task on CodeParrot and one for our Crammed BERT (Geiping & Goldstein, 2023) masked language modelling experiments pretrained on the Pile dataset (Gao et al., 2020) and fine-tuned to downstream GLUE benchmark (Wang et al., 2019). To avoid repetition, any details that are mentioned in one subsection but not the other are shared between both subsections. All runtime results on CodeParrot were run on a single A5000 GPU.

### D.1    CODEPARROT NEXT-TOKEN PREDICTION

As mentioned, much of our setup is derived from `https://huggingface.co/learn/nlp-course/chapter7/6`.

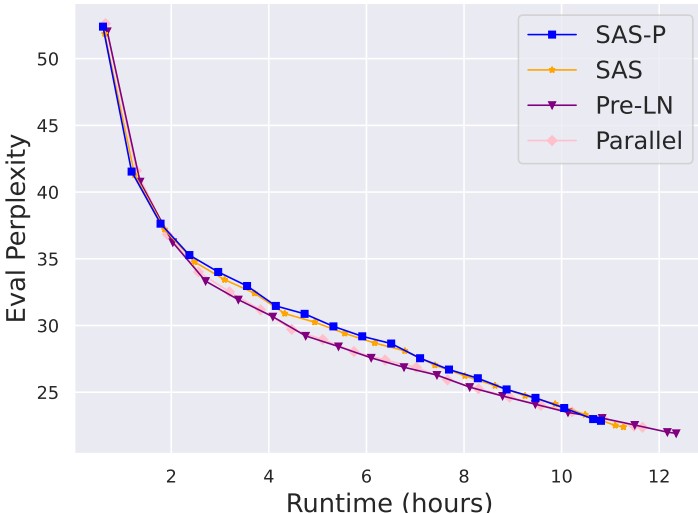

Figure 28: Eval perplexity vs runtime on an autoregressive language modelling task.

**Model** The model is a 18-layer GPT-style auto-regressive decoder-only transformer. We use width $d = 768$, and $H = 12$ heads in multi-head attention. We remove dropout entirely as our focus is on training speed, and we are always in a single-epoch regime so regularisation hurts training speed. The MLP uses ReLU activation unless stated otherwise, and we use MLP hidden dimension $3072 = 4d$. The only exception to this is in Fig. 6, where we reduce the MLP hidden dimension to $1536 = 2d$ to account for the increased memory requirements of larger depths.

For any of our simplified model we initialise $\beta_{\text{FF}} = 0.1$ in Eq. (1) to account for the lack of skip, apart from the 18-layer models in Fig. 6, where $\beta_{\text{FF}} = 0.2$ due to the narrower width.

We use RMSNorm (Zhang & Sennrich, 2019) where applicable with epsilon $1e - 8$, and add a final normalisation after the decoder. Sinusoidal positional encodings are used and added at the embedding level.

**Parameter Initialisation** For the Pre-LN and parallel blocks, we initialise all weights (including the embedding layer) to have standard deviation $0.02$ as is standard in GPT2 (Radford et al., 2019) and BERT (Devlin et al., 2018). This is a choice that is prevalent in the field, and is on the order of $O(\frac{1}{\sqrt{d}})$ for $d = 768$, that one would expect from signal propagation.

For our models, we always initialise $W^Q = 0$ like (He et al., 2023), which as discussed zeros the query-key dot product, and allows shaped attention to have a dominant identity component at initialisation. Also as discussed, we initialise trainable scalar parameters in shaped attention to 1 for simplicity; the same applies for the $\alpha_V, \beta_V$ we use in the first layer value parameters $\mathbf{W}_1^V$. All other scalar parameters in the attention and MLP branches $\beta_{\text{SA}}, \beta_{\text{FF}}$ (initialised to 1 and 0.1 resp.) are also trainable in our models, which we found to give a small boost in performance.

**Training** We use AdamW optimiser (Loshchilov & Hutter, 2017) with weight decay 0.1 which we tuned on a small grid, and found to work well for both baselines and our models. We do not apply weight decay to any scalar gain parameter. We clip gradients with with clipping parameter 1, and use epsilon of $1e - 8$ and default betas of $(0.9, 0.999)$ in AdamW. As discussed, we use a linear decay rate with $5\%$ of all steps used for linear warmup. The optimal learning rate was tuned in all cases, and for our best (SAS and SAS-P) models, was found to be $1e-3$, which exactly matched that of the default Pre-LN. This held true also when we scaled to 72 layers. V-SkipInit needed a lower learning rate for the depth scaling experiments ($3e-4$ and $1e-4$ for depths 18 and 72 respectively). We use batch size of 128 with microbatches of size 32.

**Dataset** The Codeparrot dataset is a large corpus of 20 million python files from GitHub. We take the dataset, pre-processing and tokeniser from `https://huggingface.co/learn/nlp-course/chapter7/6`. We use sequence length $T = 128$ throughout, and our tokeniser has $50K$ vocabulary size. Our base experiments train for around 43K steps on batch size 128 and sequence length 128 which is around 700M tokens. In Fig. 8 we scale this to 2B tokens.

**Task** The model is trained on next-token prediction using cross-entropy loss.

### D.2 BERT ENCODER-ONLY

As discussed in Sec. 5, we inherit much of our hyperparameters from the Cramming setup of Geiping & Goldstein (2023), and also base our implementation from their excellent codebase.[9] We highlight important implementation details here.

**Model** We use a 16-layer encoder only model, with width $d = 768$ and 12 heads. We use MLP width $3072 = 4d$, but now we use GLU (Dauphin et al., 2017) with GeLU activation, which essentially halves the hidden dimension. We use LayerNorm (Ba et al., 2016) for normalisation where applicable with epsilon $1e - 12$ as taken from Geiping & Goldstein (2023); we always use a final LN after all the layers. Again, we remove all dropout, and use a sequence length of $128$. We found our simplified skipless models prefered smaller MLP block scales and initialise $\beta_{\text{FF}} = 0.05$.

**Parameter Initialisation** The initialisations are identical to those in Codeparrot, and are detailed above.

**Datasets** Like Geiping & Goldstein (2023), we train on the Pile dataset (Gao et al., 2020), with a WordPiece tokeniser of vocabulary size 32768, and a sequence length of 128. Our fastest runs took around $600K$ steps with microbatch size 64 in 24 hours, which corresponds to around 5B tokens.

**Training** We again trained with AdamW optimiser, with weight decay 0.1. AdamW had hyparameters $[\beta_1, \beta_2] = [0.9, 0.98]$, and epsilon $1e - 12$. We used a microbatch of 64 (to fit on a RTX-2080Ti), and scale the batch size to reach 8192 linearly after $60\%$ of total training like in Geiping & Goldstein (2023). We use the same aggressive learning rate as Geiping & Goldstein (2023), which increase linearly to max value after 75% of all training steps, before linear decay, and tune the maximum learning rate to $3e - 3$ for our SAS and SAS-P models. This was slightly too large for the SAS-P model without normalisation, so we reduce to $2e - 3$. We inherit the clipping parameter of $0.5$ from Geiping & Goldstein (2023).

**Fine-tuning** We followed the same protocol as Geiping & Goldstein (2023), In particular, we fine-tune for 5 epochs with fixed hyperparameters across tasks. We found dropout to be important for good downstream performane (unlike during pre-training), and set dropout probability $p = 0.1$. We use batch size 32, with a maximum learning of $1.5e - 4$. We keep other hyperparameters, e.g. the choice of cosine decay and AdamW epsilon $1e - 6$, like from Geiping & Goldstein (2023).

**Task** The model is trained on the masked language modelling task with masking probability 0.25, as in Geiping & Goldstein (2023).

---

[9]`https://github.com/JonasGeiping/cramming`

