# OpenReview forum: "Simplifying Transformer Blocks"
_ICLR.cc/2024/Conference — ICLR 2024 poster_

### Official Review · Reviewer_wksT · 2023-10-31

**Soundness:** 3 good
**Presentation:** 4 excellent
**Contribution:** 3 good
**Rating:** 8
**Confidence:** 5

**Summary:**

This paper undertake a critical study of several components of the transformer block, with aim of simplifying the block. The authors utilize shaped attention (identity-preserving attention initialization), and scaling the MLP block to remove the value and projection matrices, and all the skip connections from transformers. The authors do detailed ablations of all variations/modifications and scaling factors, as well as studying the impact of layernorms and skip connections. The changes are studied on pre-training for GPT and BERT models, and downstream finetuning for BERT on GLUE. The proposed changes result in models with comparable/better performance compared to vanilla models, with improved training speed and fewer params.

**Strengths:**

1. The approach is well grounded on related work (SkipInit/Shaped-attention, residual scaling, etc.), and presents strong natural extension of them.
1. Completely removing the Value and Projection Matrices and skip connections offers strong simplifications.
1. The authors method achieves parameter saving and training speedup, while also resulting in simpler blocks. The training speedup could perhaps be further optimized with better implementation/variations.
1.  The experiments presented are thorough and well-motivated. They cover a wide range of variations/ablations, multiple different models, and multiple tasks.

**Weaknesses:**

1. The authors acknowledge that their experiments are in the range of smaller models (100M-300M). It will be interesting to see if the gains carry to larger models.
1. Most of the experiments are conducted on training for only 650M tokens. 650M is a low number of tokens, compared to even "compute-optimal" tokens suggested by chinchilla for the model sizes the authors use. It will be important to see if the proposed method falls behind on training for longer. That being said, Figure 8 was trained for 2B, and BERT ones for 5B.
1. The sequence length is somewhat unorthodox at only 128 - A smaller sequence length will "weaken" the ability of attention. Perhaps this could be why value projection matrices could be dropped? It is unclear how the proposed method will perform for longer sequence lengths.
1. No downstream evaluation of the GPT model.

[1] https://arxiv.org/abs/2203.15556

**Questions:**

1. By initializing attention $\beta$ as 1, but MLP $\beta_{FF}$ as $\frac{1}{\sqrt{L}}$, this results in the FFN blocks "contributing" $\frac{1}{\sqrt{L}}$ as much to the output as compared to the attention block, correct? If yes, Perhaps an interesting experiment could be decreasing the dimensions $4d$ of the MLP block when using the proposed SAS-P method - the motivation being that if the MLP block is not contributing as significantly, perhaps it can use fewer params. This could perhaps lead to significant speedups and parameter savings.
1. Were any downstream fine-tuning experiments conducted for GPT, like GLUE for BERT? It will help make paper stronger if downstream performance also remains similar/better.

Minor presentation suggestions (the authors need not respond to these) -
1. In Figure 7, some exponential smoothing should be applied to the MLM loss (or ideally, eval loss over multiple batches should so be plotted which will be hopefully much more stable.)
1. In Figure 6 , it is difficult to distinguish the plots as many of the lines overlap/intersect significantly. Perhaps some of the lines could be omitted, or separate chart provided in supplementary.

---

> ### Author Response · Authors · 2023-11-20
> **Thank you! Author response (1/1)**
>
> We thank Reviewer wksT for their time in reviewing our paper, and are glad to read that they found our work thorough and well-motivated. We address the raised concerns and questions in order as follows:
>
> 1. **Scale**: We are actively exploring ways to scale our methods up, though it is not straightforward for us as we have relatively few GPU resources available to us. We agree though that testing at larger scales will be interesting, and believe our depth scaling experiments (Figure 6) show promise in this regard.
>
> 2. **Training time**: Thanks for this point. Indeed, the motivation for Figure 8 (as well as our BERT experiments) was to demonstrate that our findings hold at longer training times. We agree that it would be interesting to scale training time further along with model size, as is commonly done in practice e.g. Chinchilla, LLaMa.
>
> 3. **Sequence Length**: Thanks for this point. We have run additional experiments with larger sequence lengths using the recent Languini Benchmark (https://arxiv.org/abs/2309.11197), which provides an open-source codebase and a large corpus of high-quality books data (based on books3 from the Pile) to run autoregressive language modelling pre-training experiments at various scales. In these experiments, we use a sequence length of 512, compared to 128 in our submission, and train on 1.2B tokens. We present evaluation perplexity curves and more experiment details in the anonymous link: https://docs.google.com/document/d/1SF8F3CXBQ_9xqTicGOjKC-UWBTKbZDTiT1zp3gqfD5c, and find that our conclusions carry over to this setting with longer sequence lengths.
>
> 4. **GPT downstream evaluation**: thanks for this point! Would you have any particular downstream benchmarks in mind, particularly for researchers with low compute resources who need to train from scratch (not using pre-trained models)? We note that the Languini Benchmark, designed for GPT language modelling research at various smaller scales, evaluates purely on pre-training performance on its books dataset  https://arxiv.org/abs/2309.11197.
>
> 5. **Downscaled MLP branches**: Thanks for this point. We certainly agree that finding an optimal “shape” for our blocks is an interesting direction, and it may be that the conventional MLP widths are not (compute) optimal for our simplified blocks.
>
>     Actually, the motivation behind downscaling the MLP block is primarily to replace the lost “implicit downweighting” effect that occurs in Pre-LN blocks (https://arxiv.org/abs/2002.10444). In this paper, the authors show that each layer $l$ in a Pre-LN architecture actually is already downweighted by a factor of $\frac{1}{\sqrt{l}}$. As our models do not have Pre-LN skips, we need to explicitly downweight the MLP otherwise we will suffer from signal propagation degeneracies at large depths. In “Stable ResNet” https://arxiv.org/abs/2010.12859, the authors show that scaling by $O(1/\sqrt L)$ in order to obtain a well-behaved infinite depth limit in terms of signal propagation theory, which is how we motivate this “stable” square-root scaling. We note two other ICLR 2024 submissions (https://openreview.net/forum?id=KZJehvRKGD and https://openreview.net/forum?id=17pVDnpwwl) have concurrently shown this stable scaling enables hyperparameter transfer in residual architectures scaled to large depths.
>
>     Having said that, inspired by your comment, we ran an additional experiment in our rebuttal experiments https://docs.google.com/document/d/1SF8F3CXBQ_9xqTicGOjKC-UWBTKbZDTiT1zp3gqfD5c to test different MLP widths in different blocks. We find reducing MLP widths (in both default and simplified blocks) is an alternative way to reduce parameter count like our removal of values and projections, but also at a slight loss in per-update convergence speed, which is counteracted through the increase in throughput. However, we note that reduced MLP widths in default blocks does not lead to simplified architectures, in terms of conceptual understanding/interpretability, unlike our proposals.

---

> > ### Comment · Reviewer_wksT · 2023-11-21
> > **Thank you for the response.**
> >
> > Longer Sequence Length Experiments: Thank you, extending sequence length to 512 adds more validity to the proposed method.
> >
> > MLP Size Reduction Experiments: I appreciate these experiments, particularly the finding that attention and MLP param counts are somewhat "interchangeable".
> >
> > GPT downstream evaluation: For down-stream evaluation, something that takes less compute would be few-shot prompting of models pre-trained in this paper on another dataset, example HumanEval, as done in the Codex paper (Chen, Mark, et al. "Evaluating large language models trained on code." arXiv preprint arXiv:2107.03374 (2021).)
> >
> > Thank you for the response, and I would like to maintain my previous rating of 8.

---

> ### Author Response · Authors · 2023-11-22
> **Thanks!**
>
> Thanks again for your time and engagement, and also for the reference, which we will look into incorporating in our work!

---

### Official Review · Reviewer_W1bM · 2023-11-03

**Soundness:** 3 good
**Presentation:** 4 excellent
**Contribution:** 3 good
**Rating:** 8
**Confidence:** 4

**Summary:**

This paper studies how to simplify the self-attention block in transformer models. Some modifications are made based on several interesting and empirical findings:
- This first is that, while somewhat similar to prior work, this paper points out that the skip connection in the self-attention subblock could be removed, without loss of training speed.
- The second is that the value and projection parameters in the sub-attention blocks could be fixed to the identity matrix, which reduces two parameterized matrices.
- Third is that the MLP sub-block skip connection could also be removed in the parallel MHA and MLP structure.
- The forth is that all normalization layers could be removed given that the proposed block has already taken the effect of normalization into account (to some extent?).

**Strengths:**

- Some interesting findings revealing the underlying working mechanism of the attention sub-block in the self-attention is reported. These findings, despite being empirical, also lead to a simplified self-attention block. It provides valuable insignts on how to design an efficient self-attention block. The community would like to see it.
- The paper did a good job explaining the step-by-step simplification flow and have provided sufficient reasons/experiments to discuss why.
- The paper is very well written, which is a pleasant read.

**Weaknesses:**

- Some critical designs in the proposed block are from existing literatures, e.g., the shaped attention (Noci et al. 2023) and the parallel structure (Wang & Komatsuzaki, 2021) of MHA and MLP sub-blocks.
- While the block is simplified, the benefits of the simplification is not significant. The simplified transformers only match the per-update training speed and performance of standard transformers. While being 15% faster and having 15% fewer parameters, some part of the throughput gain is due to the use of the parallel structure by (Wang & Komatsuzaki, 2021).
- The removal of normalization somewhat contradicts between the analysis and approaches. In Sec. 4.4, it says that the normalization layers are unnecessary, but it quickly comments that the normalization still has benefits and is still used in experiments. This somehow leads to a confusion of read.
- The performance of the simplified block slightly falls behind the standard Pre-LN and Parallel in Table 1, which means in practice the community may not use the proposed simplified block.

**Questions:**

Depite the weaknesses, I generally like this paper as it does provide some deep insights of the self-attention mechanism. I also have a few additional questions that expect the authors to reply:
- The simplification is mostly carried out in the context of skipless self-attention block. Why the block is preferred to be skipless? Is there any benefits? From the view of signal propagation theory, the skip connection benefits propagation, particularly for deeper models, and does not violate the philosophy of simplicity,.
- From Fig. 19, the pre-LN block performs worse when setting values and projections to identity. An intuition is provided, but I am not totally convinced by this intuition. In fact, I see the performance difference is almost negligible. It seems OK to set the values and projections to identity in the standard attention block with skip connection. This also reduces the number of parameters.
- Why is the performance SAS-P, no norm not reported? Is the training failed or the performance is significantly lower? In fact, I would suggest that the block can preserve the normalization layers to keep the consistency between analyses and experiments.

**Details Of Ethics Concerns:**

I do not find ethics issues.

---

> ### Author Response · Authors · 2023-11-20
> **Thank you! Author response (1/2)**
>
> We thank Reviewer W1bM for their time in reviewing our paper, and are pleased to see that they enjoyed reading our work. We address the raised concerns and questions in order, below:
>
> 1. **Critical designs from existing literature**: Though several of the tools (like shaped attention or the parallel block) we build on can be found in existing literature, it is not obvious how one would combine them in order to create simplified blocks. Moreover, we extend these existing works and make specific changes to them that are crucial for our best performance. For example, we use head dependent scalars (in shaped attention), or downweighting the MLP block $O(1/\sqrt{\text{depth}})$ in the parallel structure (which is needed to avoid the signal propagation degeneracies of the parallel block if skips are not present). Moreover, neither previous work considers removing residual connections or value/projection parameters.
>
>     We tend to agree with reviewer wksT on this point, who writes that “the approach is well grounded on related work (SkipInit/Shaped-attention, residual scaling, etc.), and presents strong natural extension of them.”
>
> 2. **Significance of simplifications**: Table 1 shows that in our implementation our parallel block only gains in 5% throughput, so 10% (out of 15%) of the throughput gain is through our further simplifications. In the era of large language models with exorbitant training and deployment costs, even 1 or 2 percent of throughput savings is significant.
>
>     Moreover, one of our key motivations for simplifications was to help close the gap between theory and practice in DL. Currently, theorists working in DL work towards understanding simplified architectures (due to convenience) and run the risk of missing out on crucial details that influence performance in practice as a result. Demonstrating that conceptually simpler architectures, with many unnecessary components removed e.g. without skip connections or values or projection parameters, can match performance of standard architectures is significant in our eyes, as it enables theorists to focus on the key factors that dictate performance in practice. Hopefully, an improved understanding acquired  through future theory can then feed back into motivating improved methods to train and finetune NNs in practice.
>
> 3. **Clarity over removing Normalisation**: Thank you for raising this point. We made a conscious effort to stress that in section 4.4, the redundancy of normalisation layers is only from a signal propagation theory at initialisation perspective. There are many empirical findings beyond the current theory, e.g. regarding normalisation and training dynamics, that the current theory is unable to predict, which is one of the motivations for this paper and presents opportunities for future work to improve upon the current theory. Notably, our most simplified block (in figure 1, top right), without normalisation, is able to match the training speed of Pre-LN transformers (e.g. in Figure 5) in terms of runtime, but we do observe slight decreases in per-step convergence in e.g. Figure 6.
>
>     Would the reviewer suggest any changes to the wording in particular to improve the clarity? Perhaps we can add "from an initialisation perspective" to the end of the first paragraph in section 4.4?
>
> 4. **Table 1**: The reviewer is correct in that the means in table 1 for our methods are very slightly worse. Having said that, up to statistical significance our simplified models match the downstream performance of standard blocks in Table 1, so we do not believe our models perform worse. Our main focus in this submission was on improving pre-training speed from random initialisations in simplified blocks, and we believe that the factors that allow for fast pre-training are not necessarily the same as good downstream performance (e.g. we need to use dropout regularisation and also much smaller learning rates to avoid overfitting at downstream time, whereas dropout only ever hurts pre-training speed). We followed exactly the fine-tuning protocol of Geiping et al in Crammed BERT (5 epochs, fixed hyperparameters across tasks etc) with minimal hyperparameter tuning besides learning rate. We are actively working towards improving our understanding of fine-tuning from pre-trained checkpoints in order to better test, and improve, the fine-tuning capabilities of our simplified models.

---

> ### Author Response · Authors · 2023-11-20
> **Author response (2/2)**
>
> 5. **Skipless attention blocks**: Thank you for raising this point. By removing the skip connection, the interpretation of the attention block is much clearer in that it simply adds to each token other tokens according to the attention matrix value, whereas previously there was a skip connection and the values/projection weights act as some unknown rotation/linear transformation. Moreover the signal propagation in the attention block has a new interpretation in terms of balancing how much a token attends to itself relative to others, which is explicitly controlled through the attention matrix. We believe these cleaner interpretations are muddied if there is a skip connection.
>
>     Furthermore, though we motivated simplified blocks for improved fundamental understanding of the roles of different architectural components (like skips) and efficiency gains, there have been other works that suggest specific disadvantages to residual architectures, e.g. more transferable adversarial examples (https://openreview.net/forum?id=BJlRs34Fvr), that may provide independent motivation to consider the skipless setting.
>
> 6. **Figure 19**: Setting the values and projections to be identity in the Pre-LN block can be thought of as being “morally” similar to removing the skip and increasing the diagonal components of the attention matrix. In this sense, we agree with the reviewer that explicitly removing the skip connection appears to not be a necessary simplification in the context of our work. However, this is not quite true in practice because of the normalisation layers acting on the residual branch in Pre-LN. This distinction is important and gives us a preference for the skipless setting for three reasons:
>  First, as described in response to the previous question we believe it is conceptually simpler to interpret a skipless block with identity val/proj compared to a Pre-LN block because the entire sub-block (and signal propagation) is determined through the attention matrix, whereas with the skip you have to juggle the skip branch with the diagonals vs off-diagonals of the attention matrix (as well as the non-linear normalisation layer). Secondly, we note that the Pre-LN model in Figure 19 with identity values and projections achieves worse final validation loss (1.213) than both our SAS and SAS-P models (1.178 and 1.187), which suggests that empirically there is a preference for skipless attention sub-blocks (we offered our intuitive explanation for this in the text accompanying Figure 19, but would be curious if the reviewer had an explanation they found more convincing?). Finally, it is cheaper to avoid implementing a full skip connection if possible, as one does not need to store activations of both branches in memory e.g. Ding et al 2021 https://arxiv.org/abs/2101.03697.
>
>     As an aside, regarding reduced parameter counts in Pre-LN blocks, the reviewer may be interested in an additional experiment we ran for the rebuttal in response to wksT, where we compare reduced widths in the MLP layers for different blocks, which can be found at this anonymous link https://docs.google.com/document/d/1SF8F3CXBQ_9xqTicGOjKC-UWBTKbZDTiT1zp3gqfD5c.
>
> 7. **SAS-P, no norm in Table 1**. As discussed in our submission (first paragraph of page 9), we found that “removing normalisation caused instabilities at fine-tuning time, where a small minority of sequences in some downstream GLUE datasets had NaN values in the initial forward pass from the pre-trained checkpoint”. This is despite the fact that we were able to pre-train SAS-P without normalisation from a random initialisation fine on the Pile (e.g. in Figure 7). We are actively working towards a better understanding fine-tuning in general (and the role of normalisation in fine-tuning), but leave this for the future as our primary focus in this present work is on pre-training speed.

---

> > ### Comment · Reviewer_W1bM · 2023-11-21
> > **Thank the authors for the responses.**
> >
> > The rebuttal has clarified a lot of confusion during my initial review. My concerns are mostly addressed.
> > From the responses of the authors, I have some further suggestions:
> > 1. Clearly state the boost percentage of throughput introduced by this paper in Table 1.
> > 2. Explain the reason why choosing the skipless block at the very beginning.
> > 3. Perhaps discuss a little bit more on the significance and the difference of simiplification between a theoretical side and a practical side. The analysis part look like a theoretical paper, while some results are more presented from a practical view. This somehow leads to my confusion.
> > 4. Clarity over removing Normalisation: Thank you for the clarification. I think the problem may be that the results do not showcase the value of removing normalisation. I think adding "from an initialisation perspective" may not be sufficient, but I do not have a better suggestion so far.
> >
> > Overall, this is a good paper. Thank you again for the detailed responses, and I would like to maintain my initial rating '8 - accept'.

---

> > > ### Author Response · Authors · 2023-11-22
> > > **Thanks!**
> > >
> > > Thanks again for your time and engagement! We will incorporate your further suggestions as best as possible into the next updated version of our work.

---

### Official Review · Reviewer_iboi · 2023-11-06

**Soundness:** 3 good
**Presentation:** 3 good
**Contribution:** 3 good
**Rating:** 6
**Confidence:** 3

**Summary:**

This work presents a thorough investigation into the complexity of standard transformer blocks, questioning the necessity of various components commonly included in their design. Through a combination of signal propagation theory and empirical observations, the authors propose a simplified transformer architecture that maintains performance levels while offering benefits in terms of training speed and model parameter count.
The paper challenges the conventional wisdom of transformer design by systematically evaluating the impact of removing certain elements, such as skip connections, projection or value parameters, sequential sub-blocks, and normalization layers. The experimental results, as reported, demonstrate that these simplifications do not detrimentally affect the training speed and can lead to a 15% increase in training throughput and a similar reduction in parameter count.

**Strengths:**

- As the work mentions, this is the first work which has simplified the transformer architecture with training throughput gains. Previous works which have tried to simplify transformer architectures have led to increase in training speeds.
- The paper is overall well written. The authors discuss each design choice in detail, which helps in understanding the motivation behind the simplifications proposed.
- The authors have appropriately discussed the limitations of their work.

**Weaknesses:**

- One weakness is that the results are limited to smaller models of size ~100M. It is not clear if the results would scale to bigger models.
- Moreover, the authors only consider GLUE benchmark for downstream evaluation. It would be nice to include a more comprehensive evaluation across different task categories like classification, generation or reasoning.

**Questions:**

- The authors mention that there is a Pre-LN and a Post-LN block and the Pre-LN work is more popular because Post-LN suffers from training instability. Is this phenomenon observed in the signal propagation literature only or all the current LM works also use this Pre-LN variant?
- The authors mention “our simplified transformers match the per-update training speed and performance of standard transformers, while enjoying 15% faster training throughput”. If the parameters are being reduced, why isn’t the per update speed faster? And, why is the throughput faster?
- On page 3, the authors mention skipless transformers are slower than those with skip connections. Why is that?
- Are there any settings (data or tasks) where this simplified transformer does not work as well compared to the standard transformer?
- What are the limitations of the proposed simplifications in terms of model expressiveness and capability?
- What is the size of the model which has 18 blocks and 768 hidden dimensions?
- In section 4,1 what is the motivation from going from SkipInit to ShapedAttention? Another question is since this work is adding identity addition to value and projection matrices, in what way this is simpler than the residual connection?

---

> ### Author Response · Authors · 2023-11-20
> **Thank you! Author response (1/2)**
>
> We thank Reviewer iboi for taking the time and effort to review our work! We will address the raised concerns and questions in order below:
>
> 1. **Scale**: We agree that it would be interesting to test our models at larger scales. However, we would like to point out that we actually experiment up to near 400M parameters, not 100M, in our depth scaling experiment in Figure 6. In Figure 6 we demonstrate that our methods scale well to 72 layers, which is already deep by modern standards (e.g. LLaMa-33B/70B have 60/80 layers respectively). These 72-layer models have 380M parameters (hidden_dim=768, mlp_dim=1536), compared to our other experiments at 18 layers, which are ~120-170M depending on the MLP dim (either 1536 or 3072). Being in academia, it is not easy for us to obtain the compute resources that would allow us to experiment at larger scales (e.g. in the billions of parameters); we are actively pursuing avenues that would allow us to do so.
>
> 2. **Other downstream evaluations besides GLUE benchmark**: Thanks for raising this point: would the reviewer happen to have any particular downstream benchmarks in mind, particularly those suitable for researchers with low GPU resources who need to train from scratch (not using pre-trained models)? Our focus in this work is primarily pre-training speed, though we agree that it is important in practice that these benefits translate to the fine-tuning setting. For that reason, we included experiments on the Crammed-BERT GLUE setting, which is popular in the community particularly for those on academic compute budgets.
>
> 3. **Pre-LN vs Post-LN**: To our knowledge, Pre-LN is used in all LMs these days, rather than the original Post-LN block. The following LM papers explicitly state this: LLaMa section 2.2 https://arxiv.org/abs/2302.13971, GPT2 section 2.3 https://d4mucfpksywv.cloudfront.net/better-language-models/language-models.pdf and other works which followed GPT-2 design e.g. Chinchilla/Gopher. One could argue that the parallel block (used in GPT-J, Palm, ViT-22B) is different to both, but we point out that the standard parallel block (e.g. Figure 1 of submission, bottom right) has a skip connection with normalisation within the residual branch, so can be thought of as the “Pre-LN parallel block”. The “Post-LN parallel block” would not scale to large depths due to signal propagation issues (e.g. rank collapse); we avoid these issues by explicitly down–weighting the MLP branch, as discussed in sections 4.3 and 4.4 of our submission.
>
> 4. **Questions about efficiency**: We believe there has been a misunderstanding here. When we use the term “per-update training speed” we mean “how fast the loss is decreasing per training step”. We show that our simplified transformers essentially match the Pre-LN arch in training speed per update in e.g. Figure 6 (note the x-axis is number of steps) or Figure 3 (where we plot the final eval loss, so lower is better training speed per update). Because we have fewer parameters and a simplified block, this translates to higher throughput and so we actually match or have slightly faster training speed by wall-clock e.g. in Figure 5 (note the x-axis is runtime). Thank you for raising this, we will reword “per-update training speed” to “per-update convergence speed”, which we hope is clearer.
>
> 5. **Why are skipless transformers slower than skip ones**: This is the question that we address in section 4.1 of our submission. We show that the issue lies in the value and projection matrices, and that one needs to restrict the updates to value and projection matrices (or equivalently reduce their learning rate, c.f. Appendix A), to recover this lost training speed (Fig 3). In the most extreme case, this means that we fix the values and projections to be the identity matrix to recover the lost training speed, which leads to our SAS block. We describe our intuitions for this in the second and third paragraphs after the bolded paragraph “Recovering lost training speed” in section 4.1.

---

> ### Author Response · Authors · 2023-11-20
> **Author response (2/2)**
>
> 6. **Other data/task settings**: In all the settings we have tried so far we have been able to emulate the default Pre-LN performance with our simplified blocks. We also present new autoregressive exps on language data (with longer context length of 512) in the rebuttal (in response to Reviewer wksT), finding similar conclusions. These experiments can be found at the following anonymous link https://docs.google.com/document/d/1SF8F3CXBQ_9xqTicGOjKC-UWBTKbZDTiT1zp3gqfD5c/. As you say, we need to test at larger scales in order to be sure that our findings are relevant in practice, but they definitely seem to be promising at present.
>
> 7. **Limitations of the proposed simplifications**: For normalisation, as discussed in the submission, though we are able to remove normalisation by following signal propagation principles, we observe slightly reduced training speeds (e.g. in Figure 6), which we plan to explore further in future work. Note that both of our main methods SAS and SAS-P include normalisation. By setting values and projections parameters to identity and removing skip connections, our models do not linearly project/rotate the values in the residual branch to a basis that is different to the identity projection of the skip branch, before they are summed. From our experiments and intuitions, this appears not to be a notable loss in capacity, especially given the matmul FLOPs saved by avoiding the value/projection weights. We discuss this question of expressivity loss from identity values and projections in more detail in the paragraph just before section 4.3 of our submission.
>
> 8. **Model size**: For nearly all settings our depth 18 Pre-LN model has 170M parameters, which is when it has MLP hidden width=3072. For the depth scaling experiments in Figure 6, we needed to reduce the MLP hidden width to 1536 to account for the 400M depth-72 model, and so our 18 layer model has 120M parameters. Our simplified models do not have values or projection parameters, which corresponds to around 15% reduction.
>
> 9. **RE V-SkipInit vs Shaped Attention**: both are motivated through ensuring good signal propagation when one scales the network to deep layers, and both achieve this by a dominant identity component in the attention matrix at initialisation, but with slightly different parameterisations. We tried both and chose shaped attention over v-skipinit because it provided very slightly better performance (figure 12 of submission).
>
> 10. **Simplification compared to residual connection**: We believe there has been a misunderstanding here. Both of our main methods, SAS and SAS-P, fix the values and projection matrices to be exactly identity. We added identity to the value and projection matrices in section 4.1 as a stepping stone towards removing them entirely later, because we wanted to present an intuitive progression for our simplifications. We believe that skipless blocks with identity values and projections are simpler than the residual blocks as it gives the simple interpretation of the (now removed) skip connection and signal propagation purely within the attention matrix itself, in terms of how much a token attends to itself relative to other tokens (e.g. the diagonals vs the off-diagonals of the attention matrix). On the other hand, with an explicit residual connection the value and projection weights are "projection matrices" which linearly transform/rotate the values, complicating how the skip relates to the attention mechanism.

---

### Meta-Review · Area_Chair_hfQ1 · 2023-12-05

**Metareview:**

This work studies ways to simplify the self-attention block in Transformers, which sheds light on which aspects of Attention are important for performance. They find various simplifications,  such as removing skip-connections, fixing projection matrices, using parallel-MLPs, and careful initialization. Many of these are closely related to existing proposals in the literature, but this work does careful ablations on each component.

All reviewers recommend acceptance, and note the clarity of the paper.
I recommend acceptance.

**Justification For Why Not Higher Score:**

The technical contribution is not extremely strong (being heavily based on prior works), and the final architecture is only a modest improvement in both simplicity & performance.

**Justification For Why Not Lower Score:**

It is valuable for the community to have careful studies of the importance of various components in Transformers, given their prevalence. Reviewers praised this paper for its clarity and its thorough ablations.

---

### Decision · Program_Chairs · 2024-01-16

Accept (poster)